# Accuracy Assessment of Photochemical Reflectance Index (PRI) and Chlorophyll Carotenoid Index (CCI) Derived from GCOM-C/SGLI with In Situ Data

Taiga Sasagawa [1,2,*], Tomoko Kawaguchi Akitsu [3], Reiko Ide [4], Kentaro Takagi [5], Satoru Takanashi [6], Tatsuro Nakaji [7] and Kenlo Nishida Nasahara [8]

1. Graduate School of Science and Technology, University of Tsukuba, 1-1-1 Tennoudai, Tsukuba 305-8572, Ibaraki, Japan
2. Geological Survey of Japan, National Institute of Advanced Industrial Science and Technology, Central 7, 1-1-1 Higashi, Tsukuba 305-8567, Ibaraki, Japan
3. Earth Observation Research Center, Japan Aerospace Exploration Agency, 2-1-1 Sengen, Tsukuba 305-8505, Ibaraki, Japan
4. Earth System Division, National Institute for Environmental Studies, 16-2 Onogawa, Tsukuba 305-8506, Ibaraki, Japan
5. Teshio Experimental Forest, Field Science Center for Northern Biosphere, Hokkaido University, 131 Toikanbetsu, Horonobe 098-2943, Hokkaido, Japan
6. Kansai Research Center, Forestry and Forest Products Research Institute, 68 Nagai-kyutaro, Momoyama-cho, Fushimi-ku, Kyoto 612-0855, Japan
7. Uryu Experimental Forest, Field Science Center for Northern Biosphere, Hokkaido University, Moshiri, Horokanai, Uryu 074-0741, Hokkaido, Japan
8. Faculty of Life and Environmental Sciences, University of Tsukuba, 1-1-1 Tennoudai, Tsukuba 305-8572, Ibaraki, Japan
* Correspondence: sasagawataiga.ryuiki@gmail.com; Tel.: +81-29-853-6705

**Abstract:** The photochemical reflectance index (PRI) and the chlorophyll carotenoid index (CCI) are carotenoid-sensitive vegetation indices, which can monitor vegetation's photosynthetic activities. One unique satellite named "Global Change Observation Mission-Climate (GCOM-C)" is equipped with a sensor, "Second Generation Global Imager (SGLI)", which has the potential to frequently and simultaneously observe PRI and CCI over a wide swath. However, the observation accuracy of PRI and CCI derived from GCOM-C/SGLI remains unclear in forests. Thus, we demonstrated their accuracy assessment by comparing them with in situ data. We collected in situ spectral irradiance data at four forest sites in Japan for three years. We statistically compared satellite PRI with in situ PRI, and satellite CCI with in situ CCI. From the obtained results, the satellite PRI showed poor agreement (the best: $r = 0.294$ ($p < 0.05$)) and the satellite CCI showed good agreement (the best: $r = 0.911$ ($p < 0.001$)). The greater agreement of satellite CCI is possibly because satellite CCI contained fewer outliers and satellite CCI was more resistant to small noise, compared to satellite PRI. Our results suggest that the satellite CCI is more suitable for practical use than the satellite PRI with the latest version (version 3) of GCOM-C/SGLI's products.

**Keywords:** GCOM-C/SGLI; photochemical reflectance index (PRI); chlorophyll carotenoid index (CCI); accuracy assessment; phenological eyes network (PEN)

## 1. Introduction

The photochemical reflectance index (PRI) is a narrow-band vegetation index proposed by Gamon et al. [1,2] as a proxy of vegetation's photosynthetic activities. PRI is affected by environmental factors such as soil moisture, precipitation, and air temperature [3–14]. It is defined as

$$PRI = \frac{\rho(531) - \rho(570)}{\rho(531) + \rho(570)},\tag{1}$$

where $\rho(\lambda)$ represents the reflectance at $\lambda$ nm wavelength. The PRI is sensitive to the short-term (such as diurnal) changes in the composition of xanthophyll pigments, namely, zeaxanthin, antheraxanthin, and violaxanthin [15,16]. These pigments convert from one to another in the xanthophyll cycle, and the changes are closely related to the photosynthetic activity. On the other hand, the PRI is also sensitive to the long-term (such as seasonal) changes in the ratio between chlorophyll and carotenoid pigment pools [15–18]. Various studies have indicated relationships between PRI and light use efficiency (LUE) [19,20], which is the ratio between the gross primary production (GPP) and the absorbed photosynthetic active radiation (APAR) at both the leaf scale [21–24] and the canopy scale [25–28]. Garbulsky et al. [29] and Zhang et al. [30] have conducted meta-analyses and found significant relationships between the PRI and the LUE over the various species and the different spatiotemporal scales. At the ecosystem scale, the PRI has been observed by satellites, aircrafts, or unmanned aerial vehicles equipped with hyperspectral imagers [5,31–34]. However, in these observations, the PRI can be irregularly obtained with only a narrow swath. Thus, some alternative indices, which are regularly available over a wide swath with satellites, have been developed for the ecosystem-scale observation of PRI.

One of the widely used alternative indices is the chlorophyll carotenoid index (CCI), which is also proposed by Gamon et al. [35]. The CCI is defined from the reflectance at 531 nm and the reflectance at red color regions as

$$CCI = \frac{\rho(531) - \rho(\mathrm{red})}{\rho(531) + \rho(\mathrm{red})} . \tag{2}$$

For example, at the ecosystem scale, a sensor named Moderate Resolution Imaging Spectroradiometer (MODIS) carried on Terra and Aqua satellites has been used to obtain the CCI [35,36]. Middleton et al. [37] obtained the CCI from MODIS band 11 (531 nm) and band 1 (645 nm). Drolet et al. [36] used the combination of MODIS band 11 and band 14 (678 nm) to obtain the CCI. CCI has been used for monitoring the changes in the LUE [8,38] and phenology [12,39]. Although CCI was developed as an alternative index of PRI, several studies have reported the CCI has the advantage over the PRI. Springer et al. [8] and Wong et al. [40] found that CCI is more sensitive to GPP than LUE at the leaf and canopy scale. Therefore, more detailed analysis of PRI and CCI based on the regular and simultaneous observation of them at the ecosystem scale was needed to understand photosynthetic activities and phenology of forests [41].

Currently, both the PRI and the CCI can be regularly obtained over a wide swath with a new satellite: Global Change Observation Mission-Climate (GCOM-C), launched by Japan Aerospace Exploration Agency (JAXA) on 23 December 2017 [42,43]. GCOM-C is equipped with a sensor named Second-Generation Global Imager (SGLI). GCOM-C/SGLI observes with 1150 km swath range for its visible and near-infrared (NIR) bands. The observation interval is once every two days around Japan (latitude = 35°N) with 14 days' revisit time. GCOM-C/SGLI's band5 (VN05: 529.7 nm), band6 (VN06: 566.1 nm), and band8 (VN08: 672.4 nm) can detect the reflectance at 531 nm, 570 nm, and red color regions, respectively. Hence, GCOM-C/SGLI has the potential to observe the PRI and the CCI simultaneously at the ecosystem scale. In addition, the spatial resolution of the PRI and the CCI obtained from GCOM-C/SGLI is 250 m, which is four times finer than that of the MODIS CCI [35–37].

Nevertheless, to the author's best knowledge, no study has conducted even the accuracy assessments of the PRI and the CCI yet in forests. Therefore, the purpose of this study is to conduct the accuracy assessments of the PRI and the CCI derived from GCOM-C/SGLI by comparing them with in situ observation data of forests.

## 2. Materials and Methods

### 2.1. Study Sites

The current study was conducted at four forest sites (Teshio: TSE, Takayama: TKY, Fuji Yoshida: FJY, Fuji Hokuroku: FHK) in Japan between 2018 and 2020 (Figures 1 and 2,

Table 1). These four sites belong to AsiaFlux [44], Japanese Long Term Ecological Research Network (JaLTER) [45], and Phenological Eyes Network (PEN) [46].

TSE is located in the northern part of Hokkaido Prefecture. The dominant species of the canopy is the young Japanese larch (a hybrid of *Larix kaempferi* and *L. gmelinii*), which is a deciduous needleleaf tree. The dominant species of the understory is the dwarf bamboo (*Sasa senanensis* or *S. kurilensis*). TSE is generally covered by snow from November to April. TKY is located on the northwestern slope of Mt. Norikura, which is a part of the Hida Mountain Range. The dominant species of the canopy are a variety of Mongolian oak (*Quercus crispula*), the Japanese white birch (*Betula platyphylla* Sukatchev var. *japonica* Hara), and the Erman's birch (*B. ermanii*). All these species are deciduous broadleaf trees. The understory is dominated by the dwarf bamboo (*S. senanensis*). TKY is usually covered by snow from December to March. FJY is located about 10 km north of Mt. Fuji. The dominant species of the canopy is the red pine (*Pinus densiflora*), which is an evergreen needle leaf tree. A variety of Mongolian oak (*Q. crispula*) and the jolcham oak (*Q. serrata*) dominate the understory. FJY is occasionally covered by snow in winter. FHK is also located about 10 km north of Mt. Fuji, and FHK is approximately 1.3 km away from FJY. The canopy is dominated by the grown Japanese larch (*L. kaempferi*), and co-dominated by red pine (*P. densiflora*), *Cornus controversa*, and a variety of Mongolian oak (*Q. crispula*). The ferns (*Dryopteris crassirhizoma* or *D. expansa*) and the dwarf bamboo (*Sasamorpha borealis*) dominate the understory. FHK is also occasionally covered by snow in winter.

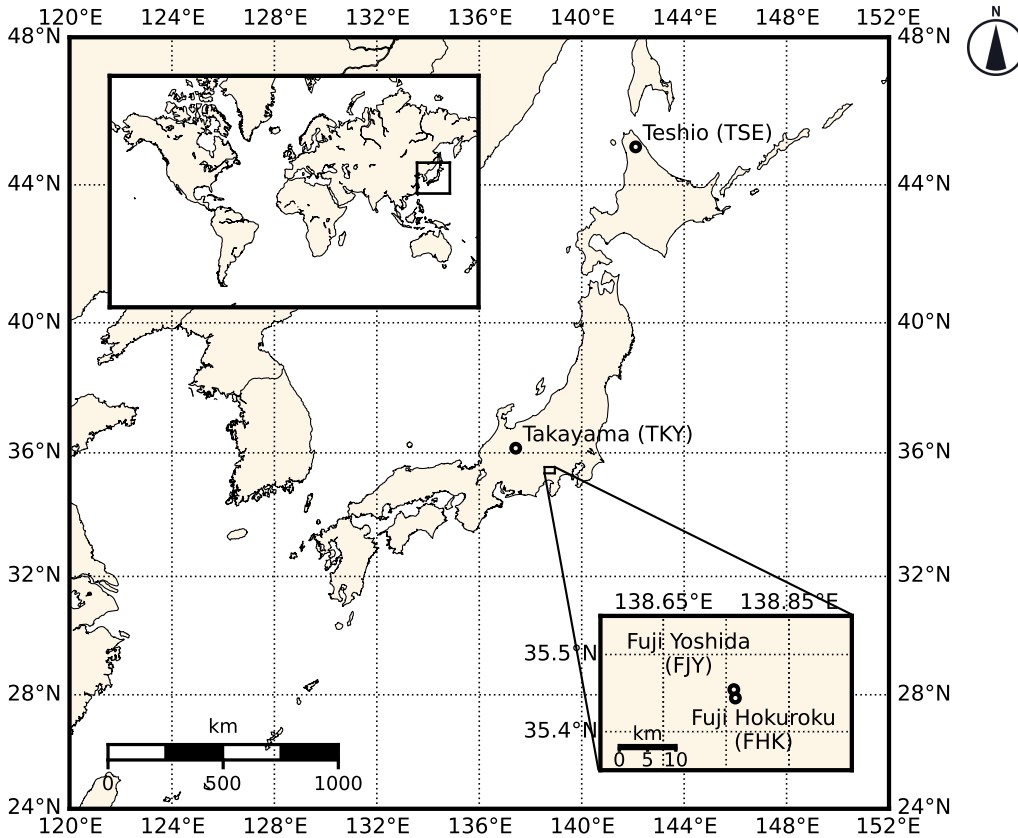

**Figure 1.** Locations of the four study sites where in situ data were collected.

## 2.2. In Situ Data

### 2.2.1. In Situ Data Collection

At each study site, we collected spectral irradiance data and fisheye images. The spectral irradiance data were used for the accuracy assessment of PRI and CCI, and the fisheye images were used for observing the conditions of vegetation, such as snow covers,

leafing, autumn colors, and leaf falling. We installed the observation instruments for spectral irradiance data and fisheye images around the top of the each observation tower (Figure 3 and Table 2).

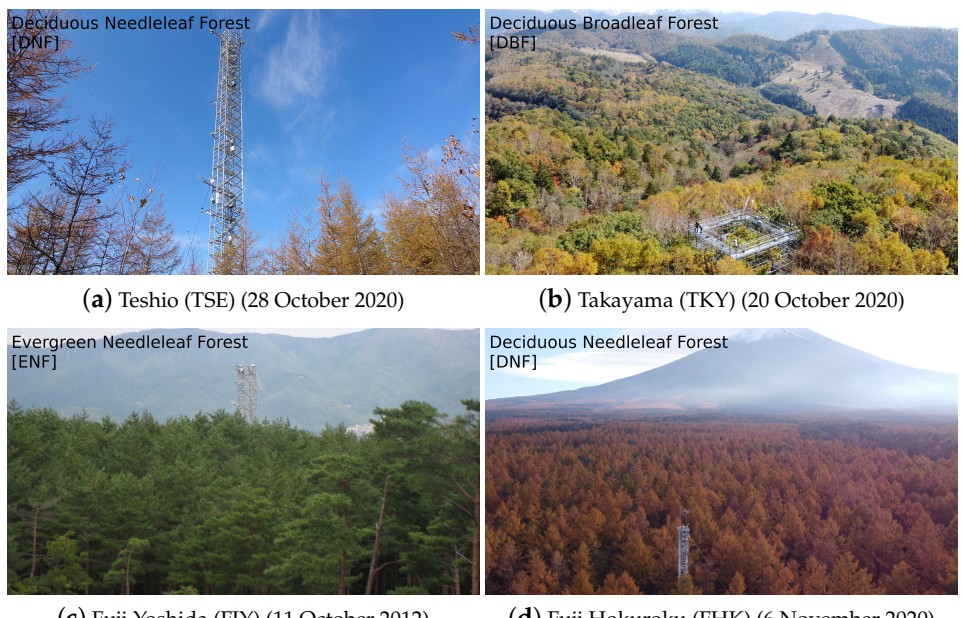

(**a**) Teshio (TSE) (28 October 2020)     (**b**) Takayama (TKY) (20 October 2020)

(**c**) Fuji Yoshida (FJY) (11 October 2012)     (**d**) Fuji Hokuroku (FHK) (6 November 2020)

**Figure 2.** Overviews of the four study sites. The silvery artificial structure in each photo is the observation tower. The date next to the site name indicates when we took each photo.

**Table 1.** Specifications of four study sites.

| Site ID | Site Name | Vegetation Type | Latitude, Longitude, and Elevation (WGS84) | Köppen–Geiger Climate Classification [47,48] | Canopy Height | Dominant Species |
|---|---|---|---|---|---|---|
| TSE | Teshio | Deciduous Needleleaf Forest | 45°3′20.99″N, 142°6′25.72″E, 70 m | Dfb | 10 m | Hybrid larch (*Larix kaempferi* × *L. gmelinii*), *Sasa senanensis*, and *S. kurilensis* |
| TKY | Takayama | Deciduous Broadleaf Forest | 36°8′42.79″N, 137°25′24.54″E, 1420 m | Dfb | 15–18 m | *Quercus crispula*, *Betula platyphylla* Sukatchev var. *japonica* Hara, *B. ermanii*, and *S. senanensis* |
| FJY | Fuji Yoshida | Evergreen Needleleaf Forest | 35°27′16.36″N, 138°45′44.10″E, 1030 m | Cfb | 20 m | *Pinus densiflora*, *Q. crispula*, and *Q. serrata* |
| FHK | Fuji Hokuroku | Deciduous Needleleaf Forest | 35°26′36.88″N, 138°45′52.93″E, 1100 m | Cfb | 25 m | *L. kaempferi*, *P. densiflora*, *Cornus controversa*, and *Q. crispula* |

In situ spectral irradiance data were measured with hemispherical spectroradiometers (HSSR): MS-700 (EKO Instrument Co., Ltd., Tokyo, Japan). The step of data acquisition was 3.3 nm from 350 nm to 1050 nm, and the full width at half maximum (FWHM) was 10 nm. MS-700 has been used in many studies [49–54]. At each of TSE, FJY, and FHK, two MS-700 were installed around the top of the each observation tower (Figure 3a). One was fixed

upward to measure the incident light from the sky, and the other was fixed downward to measure the reflected light from the vegetation (Figure 3a). At TKY, one MS-700 was installed, and the whole MS-700 was rotated by an external motor upward and downward by turns to measure the incident light and the reflected light (Figure 3b). The duration of one observation maneuver at TKY to measure the incident light and the reflected light was ten minutes. The MS-700 used in this study was occasionally (approximately every two years) calibrated by the manufacturer, using a National Institute of Standards and Technology (NIST) traceable halogen lamp.

At TSE, TKY, and FHK, we attached a masking device [55] to MS-700 for excluding the near-horizontal light from the sky and the reflected light from the body part of the observation towers (Figure 3). At FJY, the masking device was not installed.

We also collected fisheye images of the vegetation with time-lapse cameras named Automatic-capturing Digital Fisheye Camera (ADFC) (Figure 3). ADFC is a combined system of a digital camera (COOLPIX4300 or COOLPIX4500, Nikon Corp., Tokyo, Japan), a fisheye lens (FC-E8, Nikon Corp., Tokyo, Japan), and a waterproof housing case. Fisheye images taken by ADFC have been used in many studies [50,52,56,57]. Figure 4 shows examples of the fisheye images.

Acquisition intervals of the spectral irradiance data and fisheye images differed in the four study sites. The intervals are shown in Table 3. The spectral irradiance data and fisheye images were sometimes not collected. At TSE and FHK, spectral irradiance data were not collected in winter because we stopped and removed MS-700, being wary of damages caused by low temperature and snow. At TKY, the reflectance was not calculated in 2020 because of a malfunction of the masking device. At FJY, the fisheye images were not fully collected because of the problem of ADFC. However, FJY is closely located to FHK (see Figure 1), so the snow condition at FJY was referred to the snow condition at FHK in this study.

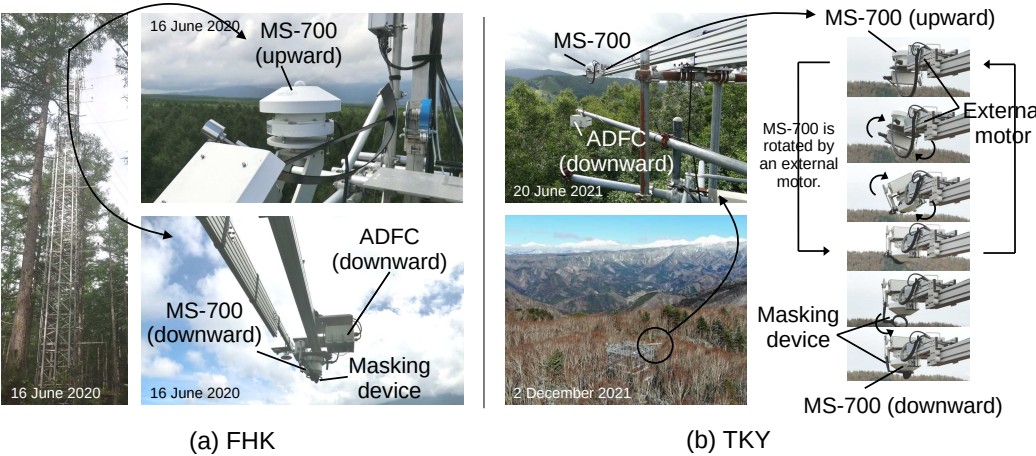

(a) FHK            (b) TKY

**Figure 3.** Examples of the instruments: MS-700, masking device, and Automatic-capturing Digital Fisheye Camera (ADFC) at (**a**) FHK and (**b**) TKY. At TSE and FJY, the instruments were installed basically in the same manner as (**a**) FHK, but FJY was not equipped with the masking device for MS-700. At TKY, an external motor rotates MS-700 to observe the incident and reflected light (**b**).

**Table 2.** The vertical positional information of the canopy and downward MS-700 at each site.

| Site ID | The Height at Where MS-700 Was Installed | The Distance between the Canopy and MS-700 |
|---|---|---|
| TSE | 23 m | 13 m |
| TKY | 18 m | 0–5 m |
| FJY | 28 m | 8 m |
| FHK | 32 m | 7 m |

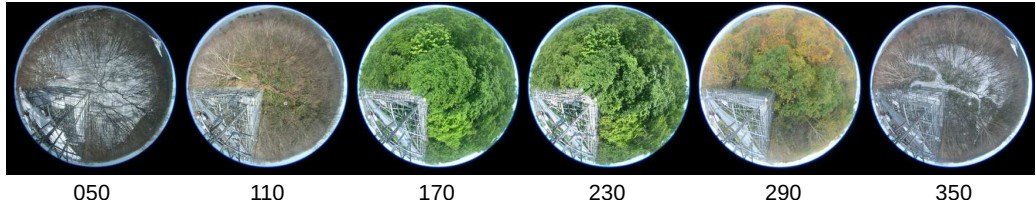

**Figure 4.** Examples of fisheye images taken by ADFC. These images were taken in 2019 at TKY. The bottom numbers represent day of year (DOY).

**Table 3.** Data acquisition intervals of MS-700 and Automatic-capturing Digital Fisheye Camera (ADFC) in the study sites. The time is described according to Japan Standard Time (JST).

| Site ID | MS-700 Upward (To the Sky) | MS-700 Downward (To the Vegetations) | ADFC Downward (To the Vegetations) |
|---------|----------------------------|--------------------------------------|-------------------------------------|
| TSE | Every 1 min 04:00–19:59 | Every 1 min 04:00–19:59 | 12:00 |
| TKY | Liner interpolation between 1 min before and 2 min after the downward observation | Every 10 min 09:10–15:00 | Every 15 min 07:00–16:45 |
| FJY | Every 10 min 04:00–20:00 | Every 10 min 04:00–20:00 | None |
| FHK | Every 2 min 06:01–18:59 | Every 4 min 06:03–18:59 | Every 1 h 06:00–18:00 |

### 2.2.2. In Situ Data Processing

From the in situ spectral irradiance data (MS-700 data), we calculated the time series of each vegetation index in two ways, namely, "original" and "simulated", for each day and each site. The "original" indices followed the original definition of PRI and CCI (Equations (1) and (2), respectively). The "simulated" indices used weighted-average reflectance within each band using GCOM-C/SGLI's relative spectral response (RSR) (see Figure 5 and Table 4). We describe their details in the followings.

To derive the original PRI ($PRI_{original}$) and the original CCI ($CCI_{original}$), the monochromatic reflectance (the reflectance in each band of MS-700) $\rho(\lambda)$ was firstly calculated as follows:

$$\rho(\lambda) = \frac{g(\lambda)}{f(\lambda)} \qquad (3)$$

where $f(\lambda)$ and $g(\lambda)$ indicate the spectral irradiance data of incident light and reflected light at $\lambda$ nm, respectively.

The spectral irradiance data precisely at 531 nm, 570 nm, and 645 nm were estimated by liner interpolation, because the spectral resolution of MS-700 was 3.3 nm and the reflectance precisely at these three wavelengths were not available. Thus, the spectral irradiance at these three wavelengths were estimated by linear interpolation. The center wavelengths of the MS-700 bands, which were the first and second closest to three wavelengths and used for linear interpolation, are listed in Table A1. $PRI_{original}$ and $CCI_{original}$ were calculated as

$$PRI_{original} = \frac{\rho(531) - \rho(570)}{\rho(531) + \rho(570)} \qquad (4)$$

and

$$CCI_{original} = \frac{\rho(531) - \rho(645)}{\rho(531) + \rho(645)}. \qquad (5)$$

To derive the simulated PRI ($PRI_{simulated}$) and the simulated CCI ($CCI_{simulated}$), we simulated bands' values of GCOM-C/SGLI as follows. First, we resampled the spectral

irradiance measured by MS-700 from 3.3 nm spectral resolution to 0.1 nm spectral resolution with linear interpolation. Then, we calculated the following values:

$$VNx_{\text{simulated}} = \frac{\int_{\Lambda} g(\lambda)\, RSR_x(\lambda)\, d\lambda}{\int_{\Lambda} f(\lambda)\, RSR_x(\lambda)\, d\lambda} \tag{6}$$

where $x$ represents the band number, $VNx_{\text{simulated}}$ represents simulated $VNx$ value, $\Lambda$ indicates the integration section (300 nm–1100 nm), $RSR_x(\lambda)$ is RSR of the band $x$, and $d\lambda = 0.1$ nm. The RSR was provided at 0.1 nm spectral resolution on the JAXA's official webpage [58]. $PRI_{\text{simulated}}$ and $CCI_{\text{simulated}}$ were calculated as:

$$PRI_{\text{simulated}} = \frac{VN05_{\text{simulated}} - VN06_{\text{simulated}}}{VN05_{\text{simulated}} + VN06_{\text{simulated}}} \tag{7}$$

and

$$CCI_{\text{simulated}} = \frac{VN05_{\text{simulated}} - VN08_{\text{simulated}}}{VN05_{\text{simulated}} + VN08_{\text{simulated}}} . \tag{8}$$

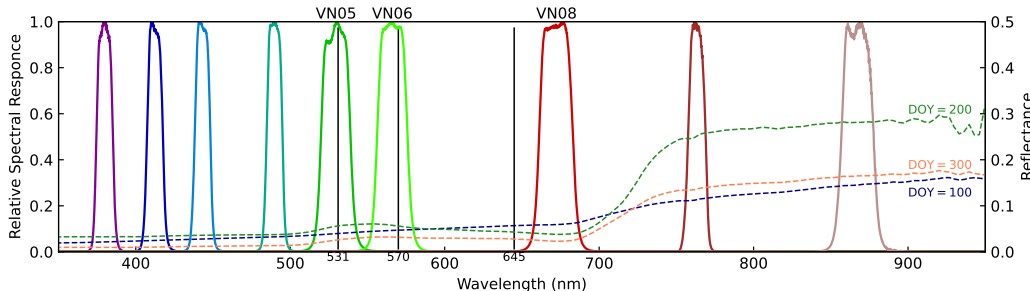

**Figure 5.** The relative spectral response (RSR) in the visible and near-infrared (NIR) range of Global Change Observation Mission-Climate (GCOM-C)/Second Generation Global Imager (SGLI). The original data was obtained from [58]. The solid black lines represent the wavelength at 531 nm, 570 nm, and 645 nm, originally used to derive the photochemical reflectance index (PRI) and the chlorophyll carotenoid index (CCI). The blue, green, and orange dotted lines are reflectance measured by MS-700 at FHK on 9 April 2020 (DOY = 100), 18 July 2020 (DOY = 200), and 26 October 2020 (DOY = 300), respectively. Each reflectance was observed at 10:31:00 (Japan standard time (JST)) on each day.

**Table 4.** Specifications of Visible and Near-Infrared Radiometer (VNR) of Second Generation Global Imager (SGLI) onboard Global Change Observation Mission-Climate (GCOM-C). The following information is based on GCOM-C data users handbook officially published by Japan Aerospace Exploration Agency (JAXA) [59].

| Band Number | Center Wavelength [nm] | Band Width [nm] | Saturation Level [W m$^{-2}$ sr$^{-1}$ μm$^{-1}$] | Instantaneous Field of View (IFOV) [m] |
|---|---|---|---|---|
| VN01 | 379.9 | 10.6 | 240–241 | 250 |
| VN02 | 412.3 | 10.3 | 305–318 | 250 |
| VN03 | 443.3 | 10.1 | 457–467 | 250 |
| VN04 | 490.0 | 10.3 | 147–150 | 250 |
| VN05 | 529.7 | 19.1 | 361–364 | 250 |
| VN06 | 566.1 | 19.8 | 95–96 | 250 |
| VN07 | 672.3 | 22.0 | 69–70 | 250 |
| VN08 | 672.4 | 21.9 | 213–217 | 250 |
| VN09 | 763.1 | 11.4 | 351–359 | 250 |
| VN10 | 867.1 | 20.9 | 37–38 | 250 |
| VN11 | 867.4 | 20.8 | 305–306 | 250 |

*2.3. Satellite Data*

2.3.1. Satellite Data Collection

GCOM-C/SGLI level 2 atmospheric corrected land surface reflectance (RSRF) daily products, whose version was 3, were used as the satellite data. The spatial resolution of the bands in visible and NIR regions of the products was 250 m. The details of each band are described in Table 4. Note that some parts of the products are not released from JAXA at present because the major update for the products (from version 2 to version 3) started in November 2021, and it has not been finished yet. Hence, the authors created the RSRF products by applying version 3 algorithm for version 2 input data: the top of atmosphere radiance (LTOA) products. Certainly, version 2 LTOA products were not the latest; however, the difference in LTOA products between version 2 and version 3 was only tiny debugs. Therefore, the RSRF products used in our study can be regarded as the same as the latest RSRF daily products, which will be freely available on JAXA's FTP server [60] (HDF5 format, WGS84 datum, sinusoidal projection). They included a quality assessment (QA) flag whose details are shown in Table 5.

**Table 5.** The details of the Quality Assessment (QA) flag of the level 2 atmospheric corrected land surface reflectance (RSRF) daily products of GCOM-C/SGLI (version 3) [61]. Bit 0 is the least significant bit.

| Bit Number | Description | Value = 0 | Value = 1 |
|---|---|---|---|
| 0 | No data | No | Yes |
| 1 | Ocean or land | Ocean | Land |
| 2 | Coast | No | Yes |
| 3 | Sun glint > 0.005 | No | Yes |
| 4 | Sun glint > 0.12 | No | Yes |
| 5 | Detection of snow or ice | No | Yes |
| 6 | Cloud by target day estimation | No | Yes |
| 7 | Probably cloud by multi day estimation | No | Yes |
| 8 | Optical thickness > 0.8 | No | Yes |
| 9 | Saturated | No | Yes |
| 10 | The number of bidirectional reflectance factor (BRF) samples $\leq 3$ | No | Yes |
| 11 | Stray light | No | Yes |
| 12 | Shadow | No | Yes |
| 13 | Detection of cloud or thick aerosol for polarization channels | No | Yes |
| 14 | Recovery of the data with previous days observation (for non-polarization bands) | No | Yes |
| 15 | Recovery of the data with previous days observation (for polarization bands) | No | Yes |

2.3.2. Satellite Data Processing

We used the highest quality RSRF data: the RSRF data whose QA flag equaled 2 (only the bit 1 equaled 1 and the others equaled 0). The satellite PRI ($PRI_{\text{satellite}}$) and the satellite CCI ($CCI_{\text{satellite}}$) were calculated using the following equations:

$$PRI_{\text{satellite}} = \frac{VN05_{\text{satellite}} - VN06_{\text{satellite}}}{VN05_{\text{satellite}} + VN06_{\text{satellite}}} \qquad (9)$$

and

$$CCI_{\text{satellite}} = \frac{VN05_{\text{satellite}} - VN08_{\text{satellite}}}{VN05_{\text{satellite}} + VN08_{\text{satellite}}}, \qquad (10)$$

where $VNx_{\text{satellite}}$ represents the reflectance value of RSRF products of the band $x$. We calculated $PRI_{\text{satellite}}$ and $CCI_{\text{satellite}}$ from the RSRF products with a Python package named "h5py" (version 2.10.0). The data processing tools developed by the authors are freely available on GitHub [62]. For the accuracy assessment of $PRI_{\text{satellite}}$ and $CCI_{\text{satellite}}$, the pixel whose center was the nearest to each site location was extracted from the RSRF products. To select the nearest pixel, we reprojected the RSRF products from sinusoidal projection to

the equirectangular projection based on the GCOM-C/SGLI manual [59]. The location of each study site and the nearest pixel were displayed in Figure 6.

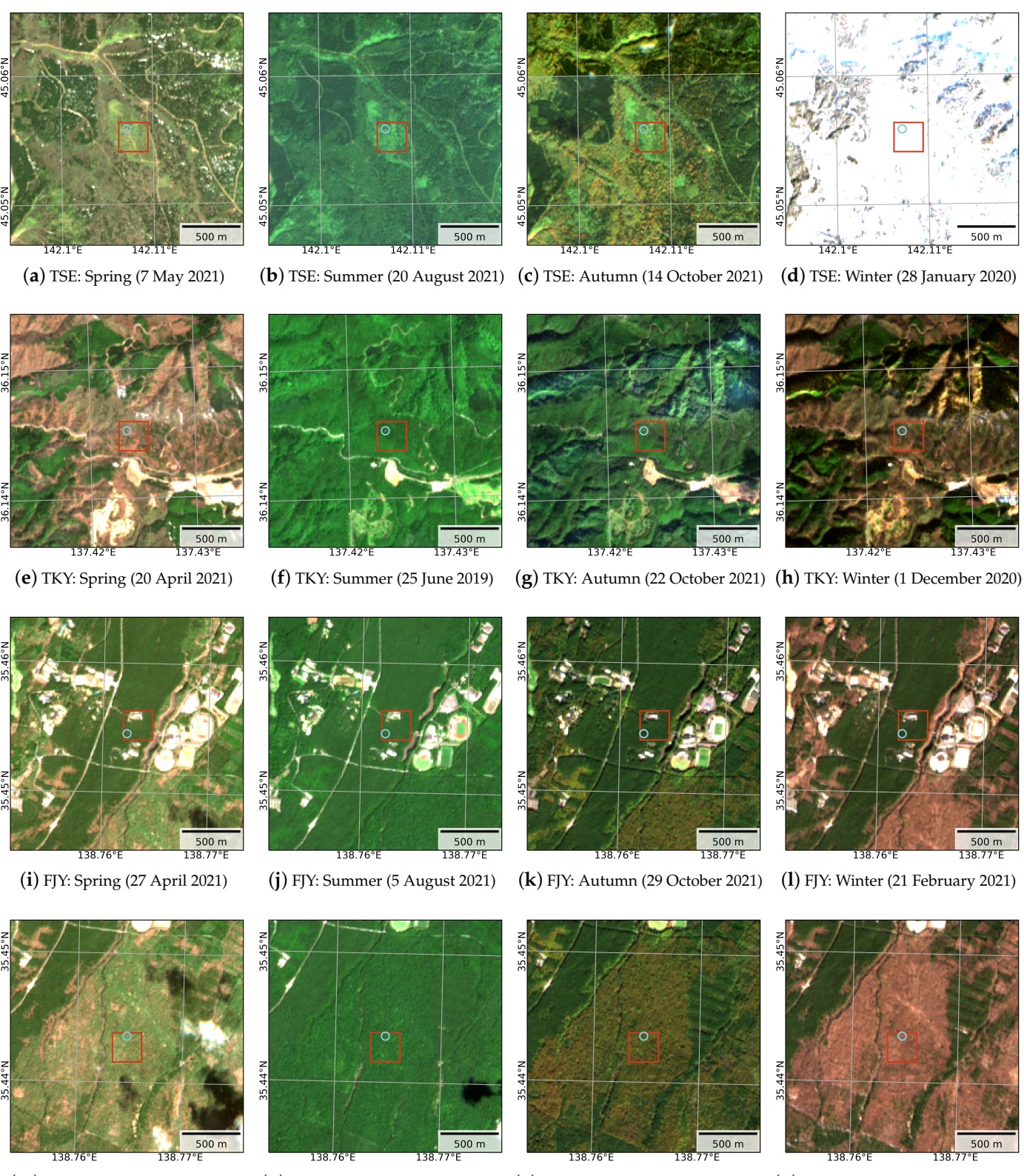

(**a**) TSE: Spring (7 May 2021)  (**b**) TSE: Summer (20 August 2021)  (**c**) TSE: Autumn (14 October 2021)  (**d**) TSE: Winter (28 January 2020)

(**e**) TKY: Spring (20 April 2021)  (**f**) TKY: Summer (25 June 2019)  (**g**) TKY: Autumn (22 October 2021)  (**h**) TKY: Winter (1 December 2020)

(**i**) FJY: Spring (27 April 2021)  (**j**) FJY: Summer (5 August 2021)  (**k**) FJY: Autumn (29 October 2021)  (**l**) FJY: Winter (21 February 2021)

(**m**) FHK: Spring (27 April 2021)  (**n**) FHK: Summer (5 August 2021)  (**o**) FHK: Autumn (29 October 2021)  (**p**) FHK: Winter (21 February 2021)

**Figure 6.** The location of each study site and the nearest pixel of GCOM-C/SGLI. The background true color image was created from Sentinel-2 level 2 products. The cyan circle indicates the location of the observation tower. The red square indicates a range of the nearest pixel of GCOM-C/SGLI.



### 2.4. Comparison and Statistical Analysis

For the sake of comparison and accuracy assessment, we extracted the original and simulated indices measured with MS-700 at the closest time to the observation time of GCOM-C/SGLI for each day and each site from their daily time series data calculated in Section 2.2.2. The observation time of GCOM-C/SGLI was obtained from the RSRF products.

Then, we displayed and compared the seasonal changes in the original, simulated, and satellite indices. The vegetation's conditions (snow covers, leafing, autumn colors, and leaf falling) were interpreted from the downward fisheye images taken by ADFC, and the conditions were simultaneously displayed. In addition, we examined the agreement between the simulated indices and the satellite indices with the scatter plots and statistics. We calculated the coefficient of correlation ($r$) , root mean square error ($RMSE$), and mean absolute error ($MAE$) as follows:

$$r = \frac{\sum_{i=1}^{n}(x_i - \bar{x})(y_i - \bar{y})}{\sqrt{\sum_{i=1}^{n}(x_i - \bar{x})^2 \sum_{i=1}^{n}(y_i - \bar{y})^2}} \, , \tag{11}$$

$$RMSE = \sqrt{\frac{1}{n}\sum_{i=1}^{n}(x_i - y_i)^2} \, , \tag{12}$$

and

$$MAE = \frac{1}{n}\sum_{i=1}^{n}|x_i - y_i| \tag{13}$$

where $n$ represents the sample size; $x_i$ and $y_i$ indicate the $i$th sample data of simulated index and satellite index; $\bar{x}$ and $\bar{y}$ were their sample means, respectively.

## 3. Results

### 3.1. Accuracy Assessment of PRI

Focusing on $PRI_{\text{original}}$ and $PRI_{\text{simulated}}$, Figure 7 shows similar seasonal trends between them. During the growing seasons (from leafing to leaf falling), however, $PRI_{\text{simulated}}$ showed a smaller range of seasonal variation than $PRI_{\text{original}}$. This was because they were approximately the same around the leafing and autumn colors, but $PRI_{\text{simulated}}$ was smaller than $PRI_{\text{original}}$ at their peak (around DOY = 200).

As for $PRI_{\text{satellite}}$, Figures 7 and 8 showed that $PRI_{\text{satellite}}$ had some significant errors (hereinafter called "outliers"), even if data screening with the QA flag, which can extract satellite data in the best condition, was applied. The outliers occurred in all four study sites, providing poor agreement.

In addition to the outliers, the relatively tiny errors of $PRI_{\text{satellite}}$, which fluctuated around the $PRI_{\text{simulated}}$ (hereinafter called "small noise"), also affected the results of the accuracy assessments. For example, $PRI_{\text{satellite}}$ in 2018 at TSE fairly matched with $PRI_{\text{simulated}}$ (Figure 7a); however, the scatter plot, including 2018, 2019, and 2020, shows the range of small noise in $PRI_{\text{satellite}}$ was approximately the same as the range of seasonal variations in $PRI_{\text{simulated}}$ (Figure 8a). Due to the small noise in $PRI_{\text{satellite}}$, the features of seasonal variations in PRI became unclear.

Therefore, owing to the outliers and small noise, the results of the accuracy assessments were not as good as expected. The best agreement between $PRI_{\text{simulated}}$ and $PRI_{\text{satellite}}$ was 0.294 ($p < 0.05$) at FHK (Table 6).

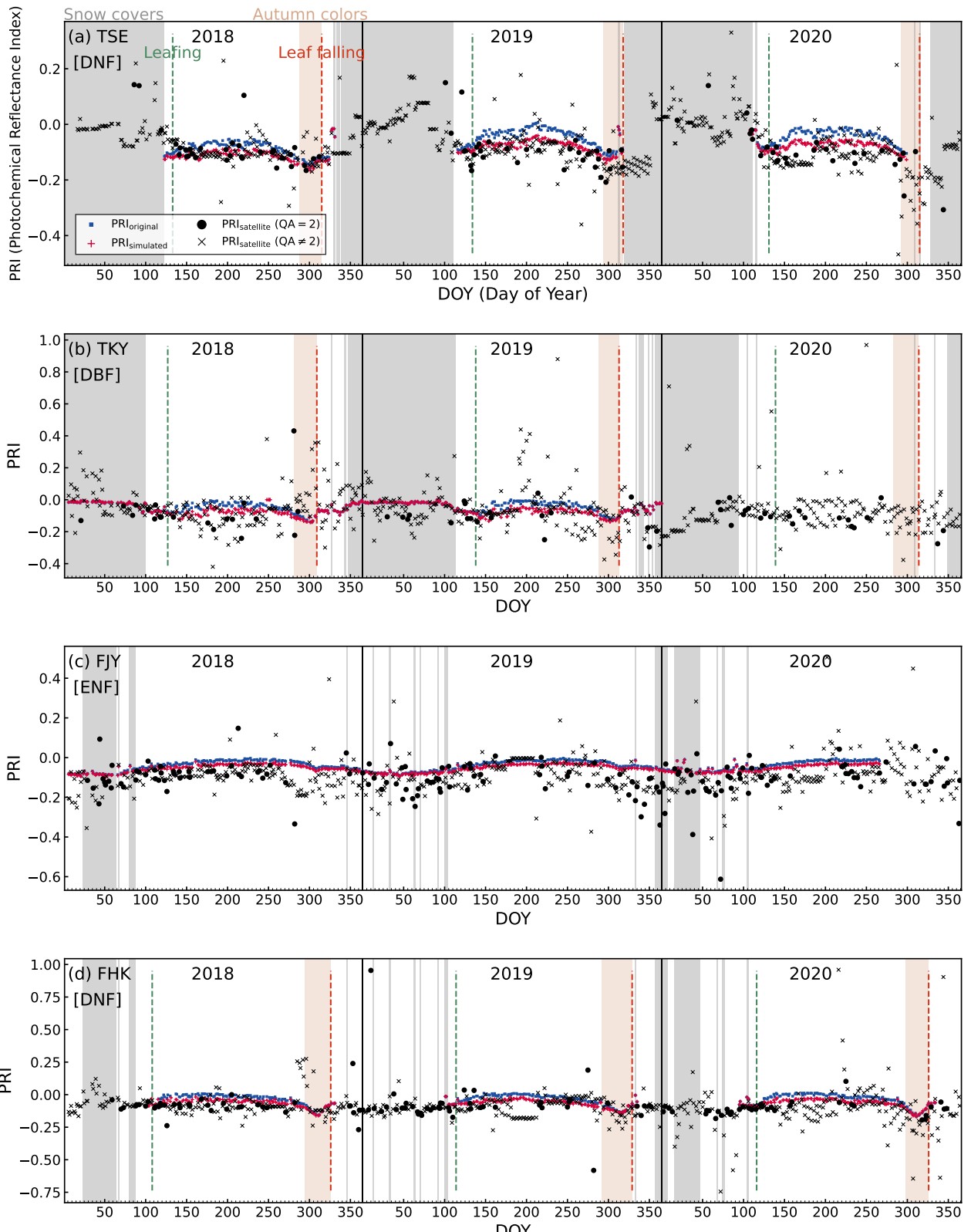

**Figure 7.** Time series of $PRI_{original}$, $PRI_{simulated}$, and $PRI_{satellite}$ from 2018 to 2020 at the four study sites. The blue square is $PRI_{original}$, the red plus is $PRI_{simulated}$, the black circle is $PRI_{satellite}$ not screened with the quality assessment (QA) flag, and the black cross is $PRI_{original}$ screened with the QA flag. The gray bands are snow seasons, the orange bands are autumn colors seasons, the green dotted lines are leafing timings, and the red dotted lines are leaf falling timings.

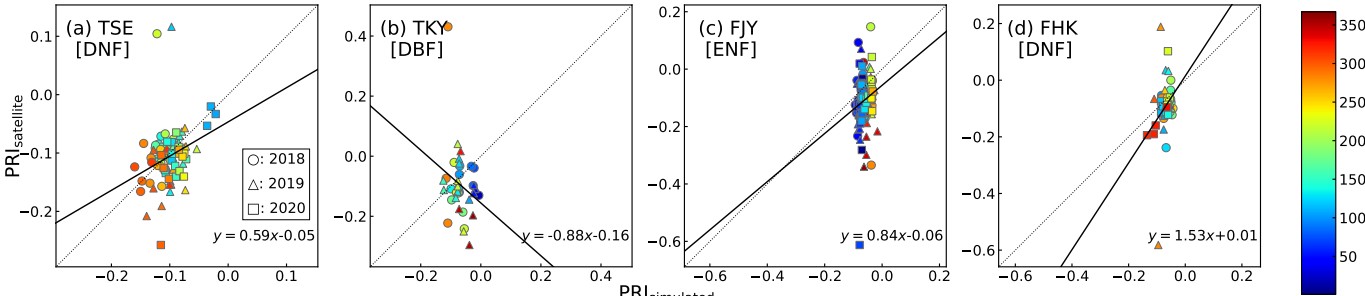

**Figure 8.** Scatter plots between $PRI_{satellite}$ and $PRI_{simulated}$. $PRI_{satellite}$ was screened with the QA flag (QA = 2). The dotted line represents the 1:1 line. The black solid line is the linear regression line. The shape of each point represents the year: the circle is 2018, the triangle is 2019, and the square is 2020. The color of each point corresponds to the DOY.

**Table 6.** The statistics of the accuracy assessment between $PRI_{satellite}$ and $PRI_{simulated}$. $n$ is the sample size, $r$ is the coefficient of correlation, $RMSE$ is the root mean square error, and $MAE$ is the mean absolute error.

| Site ID | $n$ | $r$ | $RMSE$ | $MAE$ |
|---------|-----|-----|--------|-------|
| TSE | 84 | 0.289 ($p = 0.01$) | 0.048 | 0.031 |
| TKY | 40 | $-0.245$ ($p = 0.129$) | 0.124 | 0.084 |
| FJY | 146 | 0.180 ($p < 0.05$) | 0.093 | 0.066 |
| FHK | 65 | 0.294 ($p < 0.05$) | 0.085 | 0.049 |

### 3.2. Accuracy Assessment of CCI

Figure 9 shows that $CCI_{simulated}$ had roughly similar seasonal trends to $CCI_{original}$. However, in detail, there were slight differences between them. The range of seasonal variation in $CCI_{simulated}$ was larger than that of $CCI_{original}$. At TSE and TKY, they were approximately the same around the leafing and the leaf falling seasons, but around their peak, $CCI_{simulated}$ was larger than $CCI_{original}$. At FJY, $CCI_{simulated}$ was slightly higher than $CCI_{original}$ except for the beginning of winter. At FHK, they were almost the same from the leafing to the middle of summer, and then $CCI_{simulated}$ was larger than $CCI_{original}$. Around the autumn colors season, $CCI_{simulated}$ was smaller than $CCI_{original}$. Taking together, we found that $CCI_{simulated}$ showed slightly larger seasonal variation than $CCI_{original}$ because the wavelength at the red region band of $CCI_{simulated}$ was longer than that of $CCI_{original}$ (see Figure 5).

About the $CCI_{satellite}$, Figure 9 shows that $CCI_{satellite}$ contained many outliers without the QA flag, but most of them were eliminated with the QA flag. Small noise was also contaminated with $CCI_{satellite}$, but the seasonal trends of $CCI_{satellite}$ were clear even with small noise. The reason was that the range of seasonal variation of CCI was relatively large and it was not strongly affected by the small noise (Figure 10).

Especially at TKY, there were few outliers and small noise, resulting in the best agreement between $CCI_{simulated}$ and $CCI_{satellite}$: $r = 0.911$ ($p < 0.001$), $RMSE = 0.079$, and $MAE = 0.058$ (Table 7). At both FHK and TSE, which is the deciduous needleleaf forest, $CCI_{satellite}$ reasonably matched with $CCI_{simulated}$, but $CCI_{satellite}$ showed positive bias at TSE (Figure 10a). At FJY, the coefficient of correlation was relatively low, and the relationships between $CCI_{simulated}$ and $CCI_{satellite}$ were seasonally changed; $CCI_{satellite}$ underestimated $CCI_{simulated}$ around winter but overestimated $CCI_{simulated}$ around summer. Overall, the results of the accuracy assessment of $CCI_{satellite}$ were better than those of $PRI_{satellite}$.

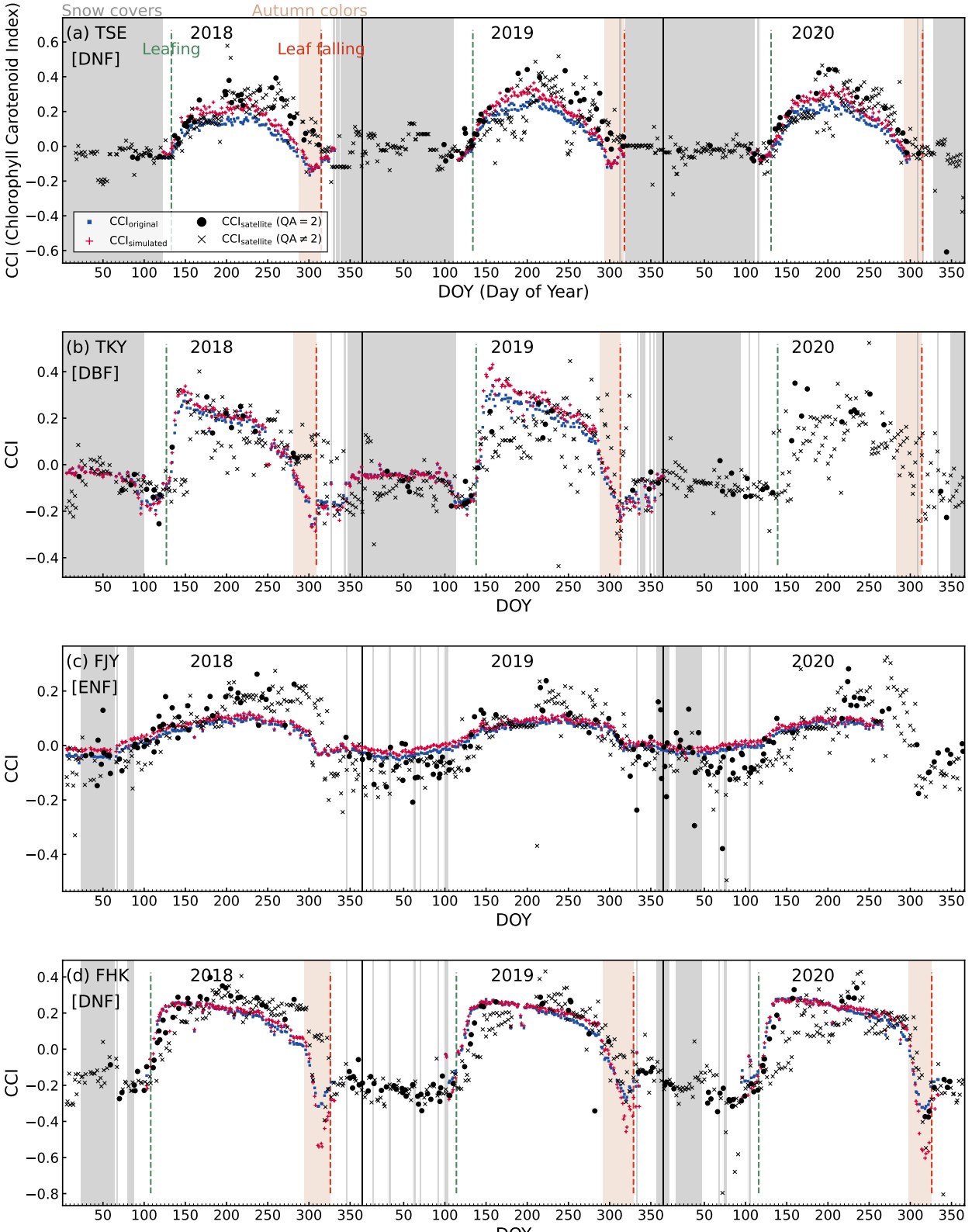

**Figure 9.** Time series of $CCI_{original}$, $CCI_{simulated}$, and $CCI_{satellite}$ from 2018 to 2020 at the four study sites. The blue square is $CCI_{original}$, the red plus is $CCI_{simulated}$, the black circle is $CCI_{satellite}$ not screened with the QA flag, and the black cross is $CCI_{original}$ screened with the QA flag. The gray bands are snow seasons, the orange bands are autumn colors seasons, the green dotted lines are leafing timings, and the red dotted lines are leaf falling timings.

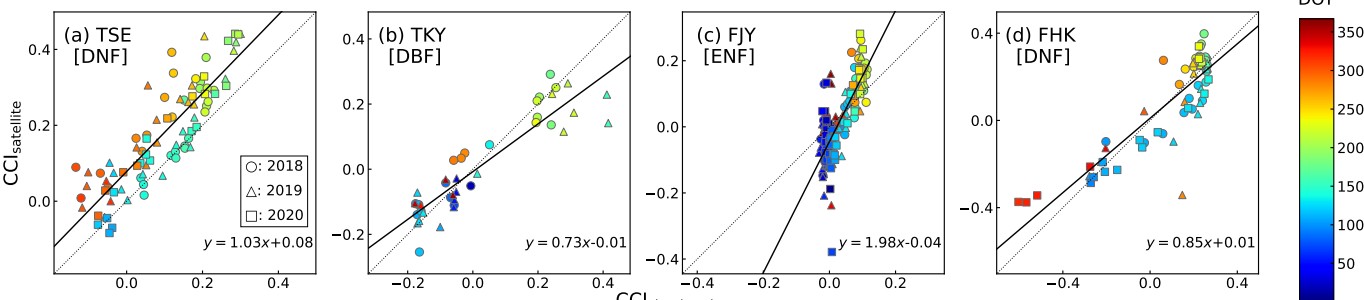

**Figure 10.** Scatter plots between $CCI_{satellite}$ and $CCI_{simulated}$. $CCI_{satellite}$ was screened with the QA flag (QA = 2). The dotted line represents the 1 : 1 line. The black solid line is the linear regression line. The shape of each point represents the year: the circle is 2018, the triangle is 2019, and the square is 2020. The color of each point corresponds to the DOY.

**Table 7.** The statistics of the accuracy assessment between $CCI_{satellite}$ and $CCI_{simulated}$.

| Site ID | *n* | *r* | *RMSE* | *MAE* |
|---|---|---|---|---|
| TSE | 84 | 0.868 ($p < 0.001$) | 0.106 | 0.086 |
| TKY | 40 | 0.911 ($p < 0.001$) | 0.079 | 0.058 |
| FJY | 146 | 0.775 ($p < 0.001$) | 0.084 | 0.065 |
| FHK | 65 | 0.874 ($p < 0.001$) | 0.112 | 0.083 |

## 4. Discussion

The PRI and the CCI are carotenoid-sensitive vegetation indices derived from the narrow green band reflectance [1,2,35–37]. The two indices are considered practical tools for monitoring the photosynthetic activities of vegetation [29,30,39–41]. According to the specifications of GCOM-C/SGLI, the satellite has the potential to derive the PRI and the CCI. Therefore, this study validated the accuracy of the satellite PRI and CCI derived from GCOM-C/SGLI by comparing them with in situ spectral data at four forest sites. As a result, we found that $PRI_{satellite}$ was poorly matched with $PRI_{simulated}$ (the best: $r = 0.294$ ($p < 0.05$) at FHK), and by contrast, $CCI_{satellite}$ matched well with $CCI_{simulated}$ (the best: $r = 0.911$ ($p < 0.001$) at TKY).

Compared to $CCI_{satellite}$, $PRI_{satellite}$ was more strongly affected by outliers and small noise, which contributed to the remarkable differences in the agreements between $PRI_{satellite}$ and $CCI_{satellite}$. Therefore, in the following section, we will discuss the reasons for the outliers and small noise. Then, we will discuss the differences in the agreements of $CCI_{satellite}$ and $CCI_{simulated}$ for each site, focusing on the footprint of the sensors and forest structures of the study sites because the agreements between $CCI_{satellite}$ and $CCI_{simulated}$ differed for each study site.

### 4.1. Common Outliers for $PRI_{satellite}$ and $CCI_{satellite}$

Some outliers were found in both $PRI_{satellite}$ and $CCI_{satellite}$ on the same day. Figures 7 and 9 showed that the common outliers sometimes occurred in snow seasons. These outliers must not be correctly removed by data screening with the QA flag (especially bit 5 for detecting snow or ice (see Table 5)).

To investigate the effects of snow contamination on $PRI_{satellite}$ and $CCI_{satellite}$, we extracted snow-free $PRI_{satellite}$ and $CCI_{satellite}$, and created the scatter plots with statistical analysis. We manually removed the $PRI_{satellite}$ and $CCI_{satellite}$ observed during the snow seasons (the gray bands in Figures 7 and 9), and one day before and after the snow seasons by checking the downward fisheye images taken by ADFC. Figure 11 shows the scatter plots of the snow-free PRI and CCI. At FJY, where the in situ data were continuously collected even in winter, the positive outliers of $PRI_{satellite}$ and $CCI_{satellite}$ around the winter seasons were successfully eliminated (Figure 11c,g). In terms of CCI, the seasonal dependency

of the relationships between $CCI_{satellite}$ and $CCI_{simulated}$ at FJY became clear. As for the statistical analysis, the correlation coefficient at FJY was improved for both PRI and CCI (Table 8). Nevertheless, at three other sites, the agreements between simulated and satellite indices were not substantially improved. The results suggest that we still need to remove the outliers which occurred in non-snow seasons.

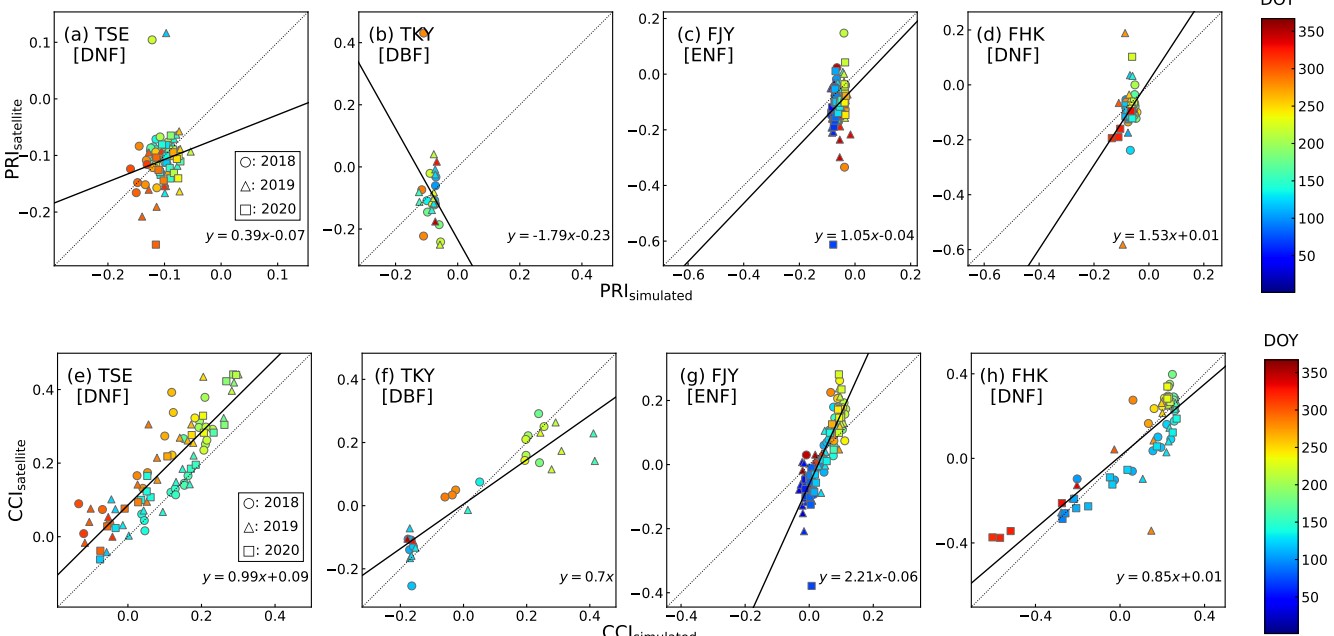

**Figure 11.** Scatter plots of snow-free PRI (**a–d**) and snow-free CCI (**e–h**). Satellite data were screened with the QA flag (QA = 2). Additionally, satellite data observed in snow seasons were manually removed. The dotted line represents the 1 : 1 line. The black solid line is the linear regression line. The shape of each point represents the year. The color of each point corresponds to the DOY.

**Table 8.** The statistics of the accuracy assessment for snow-free $PRI_{satellite}$ and snow-free $CCI_{satellite}$.

|  | Site ID | $n$ | $r$ | RMSE | MAE |
|---|---|---|---|---|---|
| PRI | TSE | 81 | 0.166 ($p = 0.138$) | 0.049 | 0.031 |
|  | TKY | 29 | $-0.274$ ($p = 0.150$) | 0.126 | 0.078 |
|  | FJY | 115 | 0.228 ($p < 0.05$) | 0.090 | 0.063 |
|  | FHK | 65 | 0.294 ($p < 0.05$) | 0.085 | 0.049 |
| CCI | TSE | 81 | 0.863 ($p < 0.001$) | 0.108 | 0.088 |
|  | TKY | 29 | 0.915 ($p < 0.001$) | 0.089 | 0.065 |
|  | FJY | 115 | 0.847 ($p < 0.001$) | 0.084 | 0.064 |
|  | FHK | 65 | 0.874 ($p < 0.001$) | 0.112 | 0.083 |

We also found common outliers in non-snow seasons. We illustrated one example of common outliers at FHK with true color images and spatial distributions of $PRI_{satellite}$ and $CCI_{satellite}$ in Figure 12. The figure shows that the observation site was located on the boundary region between the screened and non-screened areas. In order to reveal the actual sky condition at the observation time, we displayed the sky images taken by upward ADFC around the observation time of GCOM-C/SGLI in Figure 13. The figure implies that it must be cloudy at the observation time. This finding suggests that the cloud screening with the QA flag (bit 6 and bit 7 for detecting cloud) might be insufficient, and the overlooking of the cloud might occur around the boundary region.

As Motohka et al. [50] reported, such cloud contamination is a severe problem for the analysis of vegetation indices. Therefore, the QA flag of GCOM-C/SGLI should be improved in future updates of the products.

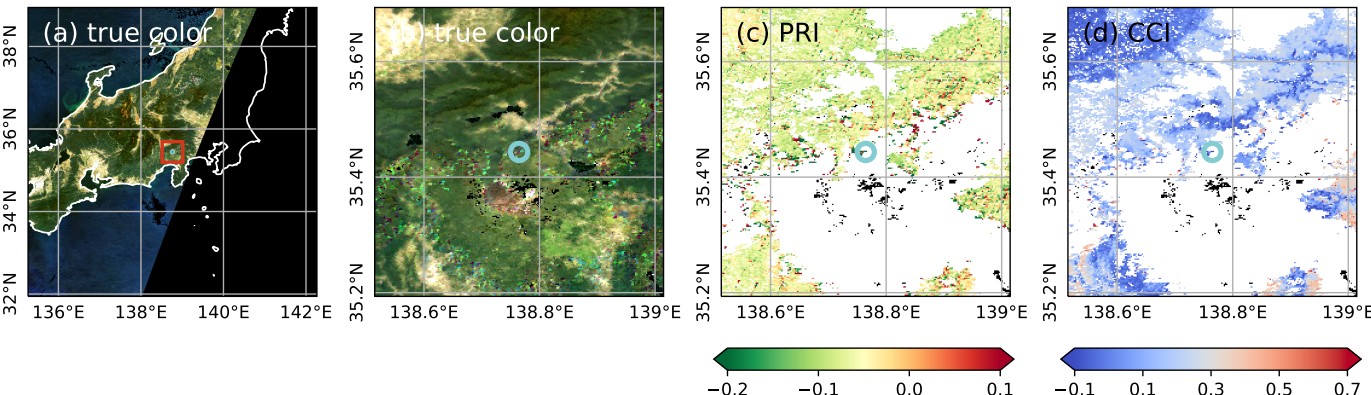

**Figure 12.** The spatial distribution of $PRI_{\text{satellite}}$ and $CCI_{\text{satellite}}$ on 9 October 2019 (DOY = 282) around FHK. The cyan circles in each figure indicate the location of FHK. The red rectangle in (**a**) represents the range of (**b**–**d**). (**a**) shows the true color image of GCOM-C/SGLI. (**b**) shows the true color image, (**c**) shows $PRI_{\text{satellite}}$, and (**d**) shows $CCI_{\text{satellite}}$. (**a**,**b**) are not screened with the QA flag and (**c**,**d**) are screened with the QA flag. The black area represents where the RSRF product was unavailable, and the white area represents the screened area with the QA flag.

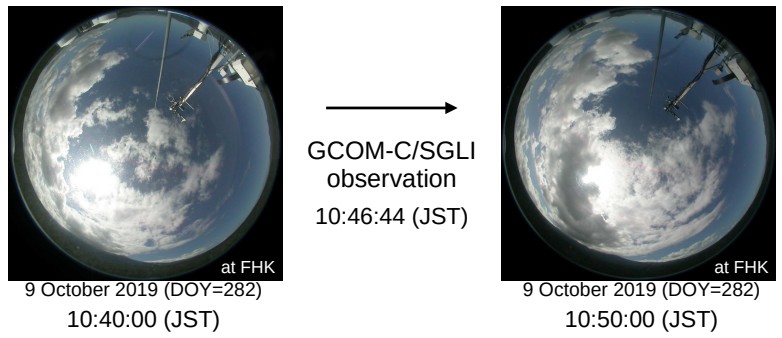

**Figure 13.** The sky images taken by upward ADFC around the observation time of GCOM-C/SGLI (10:46:44 (JST)) at FHK on 9 October 2019 (DOY = 282).

*4.2. Unique Outliers for $PRI_{\text{satellite}}$*

In addition to the common outliers for $PRI_{\text{satellite}}$ and $CCI_{\text{satellite}}$, we found two kinds of outliers unique to $PRI_{\text{satellite}}$. One is the "striping" outliers and the other is the "cluster" outliers.

### 4.2.1. Striping Outliers

One example of the striping outliers is displayed in Figure 14. Figure 14 shows that the striping outliers occurred parallel from northeast to southwest only in $PRI_{\text{satellite}}$. We also illustrated the spatial distribution of the reflectance of VN05, VN06, and VN08, which were used for calculating $PRI_{\text{satellite}}$ and $CCI_{\text{satellite}}$ in Figure 15. Figure 15 shows that no striping outliers occurred in VN05 and VN06. In short, the striping outliers must be due to the position of detectors on SGLI for each band and the nearest neighbor resampling for geometric correction of RSRF products. According to the official document of GCOM-C/SGLI [59], SGLI is equipped with the detectors of each band in a line; each detector has different relationships between the spatial coordinates on the satellite data and spatial coordinates on the Earth's surface. The detectors of VN05 and VN06 are installed relatively far from each other, whereas the detectors of VN05 and VN08 are installed next to each other. Hence, the difference in the relationships of spatial coordinates between VN05 and VN06 was larger than that between VN05 and VN08. Then, because of the geometric correction with nearest neighbor resampling, the correspondence of pixels between VN05 and VN06 must sometimes be displaced by more than one pixel. As a result, after the geometric correction, some pixels in VN05 and VN06 must refer to the different pixels in

the input data of geometric correction. This difference possibly caused the striping outliers in $PRI_{satellite}$ when $PRI_{satellite}$ was calculated. By contrast, the displacement between VN05 and VN08 must be less than one pixel. Hence, pixels in VN05 and VN08 may refer the same pixels in the input data of geometric correction, and no striping outliers was caused in $CCI_{satellite}$.

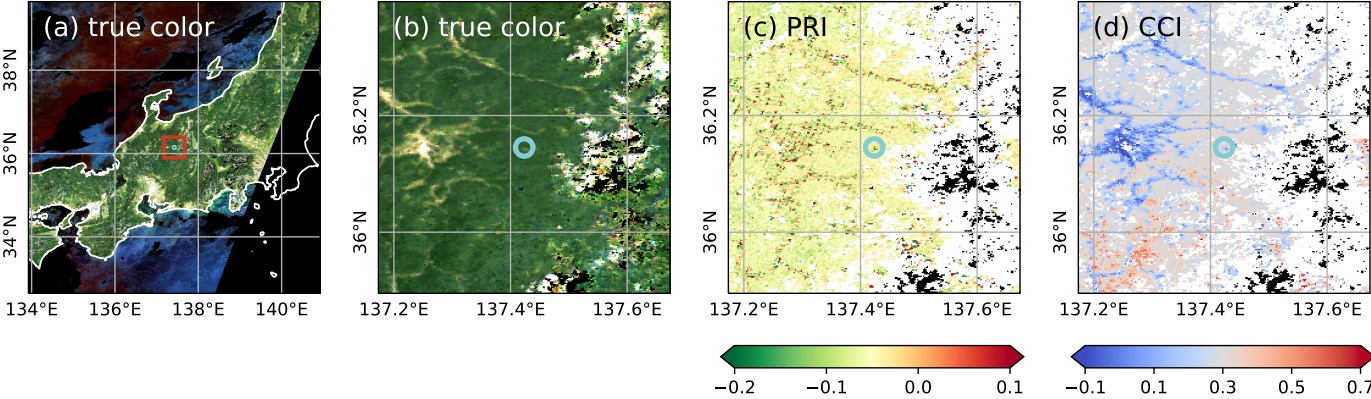

**Figure 14.** The spatial distribution of $PRI_{satellite}$ and $CCI_{satellite}$ on 2 August 2019 (DOY = 214) around TKY. The cyan circles in each figure indicate the location of TKY. The red rectangle in (**a**) represents the range of (**b**–**d**). (**a**) shows the true color image of GCOM-C/SGLI. (**b**) shows the true color image, (**c**) shows $PRI_{satellite}$, and (**d**) shows $CCI_{satellite}$. (**a**,**b**) are not screened with the QA flag and (**c**,**d**) are screened with the QA flag.

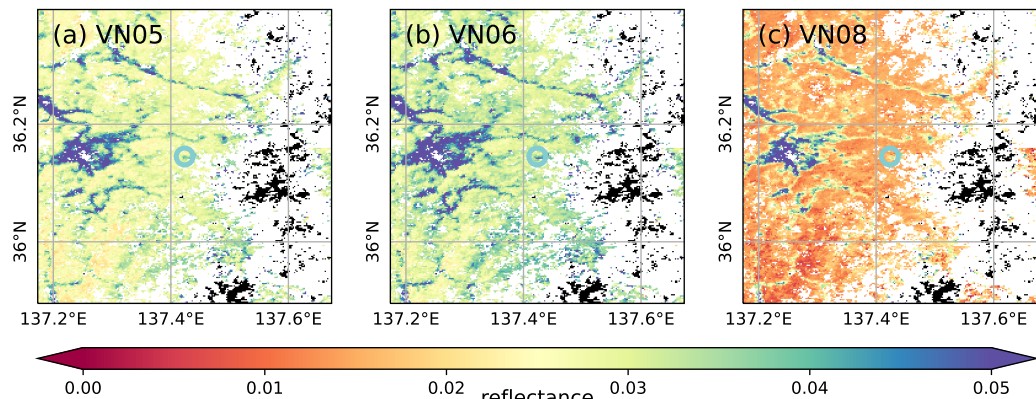

**Figure 15.** The spatial distribution of (**a**) $VN05_{satellite}$, (**b**) $VN06_{satellite}$, and (**c**) $VN08_{satellite}$ on 2 August 2019 (DOY = 214) around TKY. The range of each map is the same as that of Figure 14b–d. All maps are screened with the QA flag. The cyan circles in each figure indicate the location of TKY.

### 4.2.2. Cluster Outliers

In addition to the striping outliers, one example of the cluster outliers is illustrated in Figure 16. We found that the outliers were distributed like clusters only in $PRI_{satellite}$ (Figure 16c), but not in $CCI_{satellite}$ (Figure 16d). We also displayed the spatial distribution of the reflectance of VN05, VN06, and VN08 in Figure 17. According to Figure 17, VN06 had the cluster outliers around where $PRI_{satellite}$ contained the cluster outliers. Thus, the cluster outliers in $VN06_{satellite}$ may be one factor of the cluster outliers in $PRI_{satellite}$. As for the cluster outliers in VN06, however, we found no significant reasons. Thus, we will continue to pursue the reasons and solutions for the cluster outliers in future work.

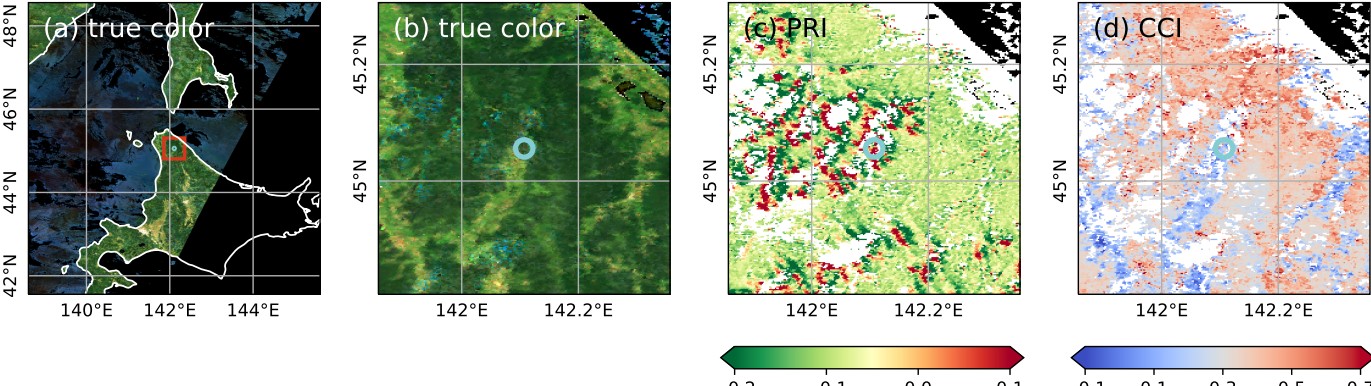

**Figure 16.** The spatial distribution of $PRI_{satellite}$ and $CCI_{satellite}$ on 8 August 2018 (DOY = 220) around TSE. The cyan circles in each figure indicate the location of TSE. The red rectangle in (**a**) represents the range of (**b–d**). (**a**) shows the true color image of GCOM-C/SGLI. (**b**) shows the true color image, (**c**) shows $PRI_{satellite}$, and (**d**) shows $CCI_{satellite}$. (**a,b**) are not screened with the QA flag and (**c,d**) are screened with the QA flag.

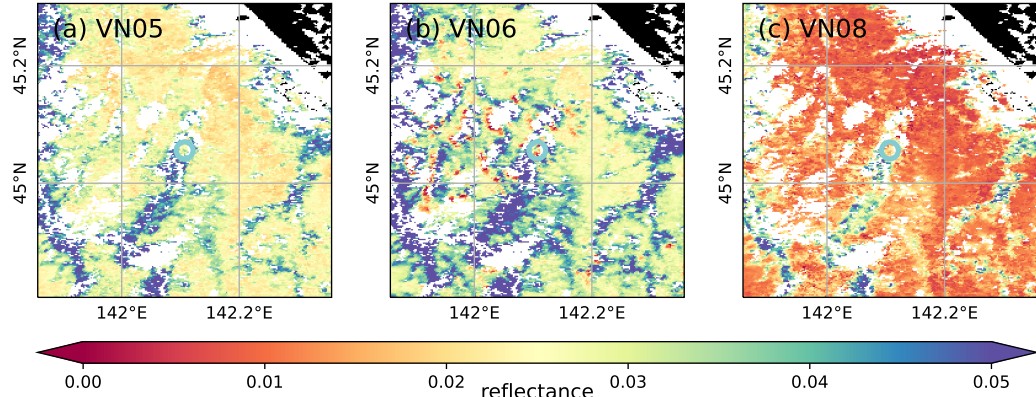

**Figure 17.** The spatial distribution of (**a**) $VN05_{satellite}$, (**b**) $VN06_{satellite}$, and (**c**) $VN08_{satellite}$ on 8 August 2018 (DOY = 220) around TSE. The range of each map is the same as that of Figure 16b–d. All maps are screened with the QA flag. The cyan circles in each figure indicate the location of TSE.

### 4.3. Demonstrations of Removing the Outliers

As we discussed in Sections 4.1 and 4.2, both $PRI_{satellite}$ and $CCI_{satellite}$ contained the common outliers, and $PRI_{satellite}$ uniquely included the striping and cluster outliers. Hence, we need to remove these outliers for the application of $PRI_{satellite}$ and $CCI_{satellite}$. Here, we propose to calculate the spatial mean as one method to remove the outliers.

According to the JAXA's official report, the uncertainty of the geometric correction was less than 0.5 pixels [63]. Thus, we calculated the mean and standard deviation of four neighbor pixels, including the nearest neighbor pixel shown in Figure 6. For the calculation of the mean and standard deviation, if at least one pixel in four neighbor pixels was screened with the QA flag, we did not calculate the mean and standard deviation for accuracy assessments.

First, we show the results of PRI in Figure 18a–d and Table 9. As a result of calculating the spatial mean, many outliers were removed, and *r*, *RMSE*, and *MAE* were improved in many cases. At TKY, the results were drastically improved, and at FJY, the outliers, which we could not remove by using snow-free data set in Section 4.4, were eliminated.

Next, the results of CCI are shown in Figure 18e–h and Table 9. The outliers of CCI were also removed, and especially at FJY and FHK, we successfully removed the significant outliers. By contrast, the statistical result at TKY was not improved. The reason may be the

spatial resolution of $CCI_{\text{satellite}}$ became too coarse, by calculating the mean value with four neighbor pixels.

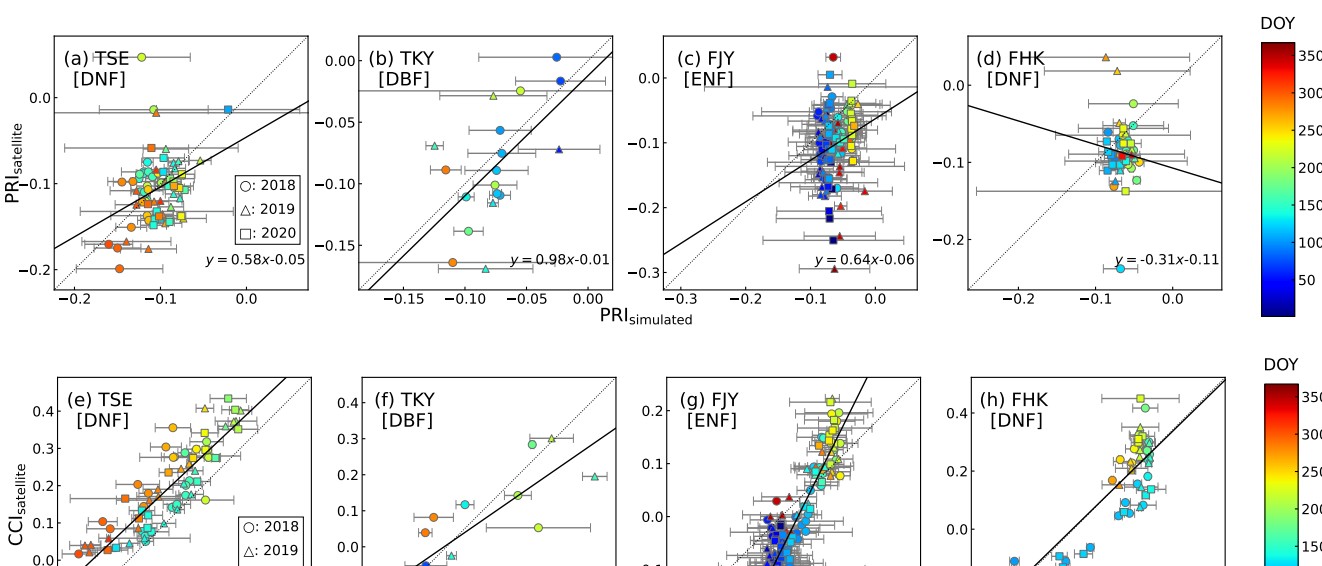

**Figure 18.** Scatter plots of spatial mean of $PRI_{\text{satellite}}$ (**a–d**) and $CCI_{\text{satellite}}$ (**e–h**) calculated from four neighbor pixels. Satellite data were screened with the QA flag (QA = 2). The error bar represents standard deviation for each point. The dotted line represents the 1 : 1 line. The black solid line is the linear regression line. The shape of each point represents the year. The color of each point corresponds to the DOY.

**Table 9.** The statistics of the accuracy assessment of $PRI_{\text{satellite}}$ and $CCI_{\text{satellite}}$ with the mean value of four neighbor pixels.

|     | Site ID | $n$ | $r$ | RMSE | MAE |
| --- | --- | --- | --- | --- | --- |
| PRI | TSE | 69 | 0.339 ($p < 0.01$) | 0.038 | 0.028 |
|     | TKY | 18 | 0.596 ($p < 0.01$) | 0.039 | 0.034 |
|     | FJY | 120 | 0.217 ($p < 0.05$) | 0.065 | 0.050 |
|     | FHK | 53 | $-0.088$ ($p = 0.531$) | 0.045 | 0.034 |
| CCI | TSE | 69 | 0.874 ($p < 0.001$) | 0.104 | 0.088 |
|     | TKY | 18 | 0.884 ($p < 0.001$) | 0.087 | 0.064 |
|     | FJY | 120 | 0.886 ($p < 0.001$) | 0.071 | 0.058 |
|     | FHK | 53 | 0.901 ($p < 0.001$) | 0.085 | 0.070 |

### 4.4. Small Noise and Index Design

In addition to the outliers, $PRI_{\text{satellite}}$ and $CCI_{\text{satellite}}$ included small noise. One of the factors of small noise can be the uncertainty of the geometric correction. JAXA reported that the uncertainty of the geometric correction in version 3 RSRF products was of less than 0.5 pixels [63]. However, uncertainty of less than 0.5 pixels can be insufficient, particularly at TSE, where the homogeneous vegetation area was limited (see Figure 6a–d). The uncertainty possibly caused the contamination of other vegetation types, which must induce the small noise.

Moreover, the uncertainty of atmospheric correction can also be responsible for the small noise. We demonstrated the accuracy assessment of VN05, VN06, and VN08 in Figure A1 and Table A2. The results showed insufficient accuracy, especially for VN05 and VN06.

The small noise affected the results of the accuracy assessment of $PRI_{\text{satellite}}$ more strongly than $CCI_{\text{satellite}}$, as seen in Figures 8 and 10. The difference in the effects of small

noise between $PRI_{\text{satellite}}$ and $CCI_{\text{satellite}}$ may be attributed to the design of each index. As can be seen in Figures 7 and 9, the range of seasonal variation in $PRI_{\text{satellite}}$ was smaller than that in $CCI_{\text{satellite}}$. Hence, $PRI_{\text{satellite}}$ was easily and strongly affected by the small noise in comparison to $CCI_{\text{satellite}}$. At the leaf and the canopy scale, such a small noise might not occur frequently and PRI worked well to monitor vegetation's conditions. However, small noise easily happened for satellite remote sensing, which requires atmospheric correction, geometric correction, and bidirectional reflectance distribution function (BRDF) correction. Therefore, compared to PRI, CCI may be more suitable for monitoring the vegetation with satellite remote sensing because CCI is "resistant" and PRI is "susceptible" to the small noise.

*4.5. Footprint Effects for Accuracy Assessment of $CCI_{\text{satellite}}$*

As listed in Tables 6 and 7, the accuracy of $CCI_{\text{satellite}}$ was higher than that of $PRI_{\text{satellite}}$ at all study sites. However, the agreements between $CCI_{\text{satellite}}$ and $CCI_{\text{simulated}}$ differed for each site. Especially at TSE and FJY, we found the distinctive relationships between $CCI_{\text{satellite}}$ and $CCI_{\text{simulated}}$ (Figure 10):

At TSE, indeed, the agreement was reasonably good ($r = 0.868$ ($p < 0.001$)), but $CCI_{\text{satellite}}$ showed a positive bias. The positive bias was probably due to the differences in the footprint between in situ sensor (MS-700) and the satellite sensor (SGLI). As mentioned in Section 2.1, the dominant species at TSE was "young" Japanese larch, and its population density at the canopy was not high, as shown in Figure 19a. Hence, around the edge of the observation area of MS-700, MS-700 laterally observe the sides of trees (Japanese larch), rather than from straight above, and the trees tend to hide the understory (dwarf bamboo) behind them. In contrast, GCOM-C/SGLI observes the trees from above. Therefore, the contribution of Japanese larch to the reflectance observed by MS-700 might be larger than that observed by the GCOM-C/SGLI, and it may cause the positive bias.

At FJY, the relationships between $CCI_{\text{satellite}}$ and $CCI_{\text{simulated}}$ were seasonally changed, as seen in Figure 10c (clearer in Figure 11g). The seasonal dependency might be affected by the difference in footprints between MS700 and SGLI. As we mentioned in Section 2.2.1, the masking device, which excludes the reflected light from the body part of the observation tower, was not installed at FJY. Thus, as seen in Figure 19c, around ten percent of the footprint of MS-700 was occupied by the observation tower, whereas the area of the observation tower in one pixel of SGLI was tiny. Then, the observation tower contributed the reflectance measured by MS-700 more significantly than that observed by SGLI. The reflected light from the observation tower can be affected by some variables, such as Sun elevation, which changes seasonally. Therefore, the relationships between $CCI_{\text{simulated}}$ and $CCI_{\text{satellite}}$ may be seasonally changed.

As discussed above, the difference in the footprint might be responsible for the distinctive relationships between $CCI_{\text{simulated}}$ and $CCI_{\text{satellite}}$ at TSE and FJY. We are aware that the difference in the footprint was the limitation of this study. In contrast, at TKY and FHK, the forests were relatively close to climax (Figure 19b,d), and the vegetation's condition in the footprint of MS-700 may be similar to that of GCOM-C/SGLI. Indeed, the results of accuracy assessment at TKY and FHK were better than those at TSE and FJY.

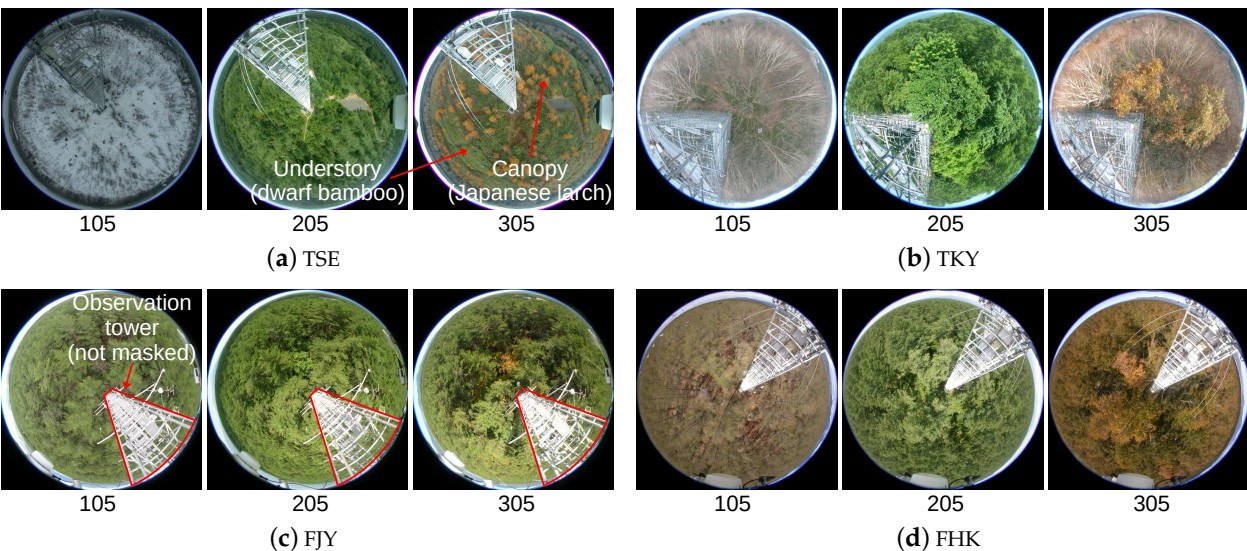

**Figure 19.** The downward fisheye images taken by ADFC in 2018 at (**a**) TSE, (**b**) TKY, (**c**) FJY, and (**d**) FHK. The numbers under the images are DOY when the images were taken. The images approximately display the approximately same observation area as downward MS-700 for each site.

## 5. Conclusions

The accuracy assessments of PRI and CCI derived from GCOM-C/SGLI were conducted by comparing them with in situ data. As a result, GCOM-C/SGLI provided $PRI_{satellite}$ with poor accuracy (the best: $r = 0.294$ ($p < 0.05$) at FHK) and $CCI_{satellite}$ with good accuracy (the best: $r = 0.911$ ($p < 0.001$) at TKY). Thus, $CCI_{satellite}$ must be suitable for monitoring photosynthetic activities of vegetation with RSRF products (version 3) of GCOM-C/SGLI rather than $PRI_{satellite}$.

We found that some outliers, possibly caused by insufficient QA flags, affected the results of the accuracy assessments of both $PRI_{satellite}$ and $CCI_{satellite}$. Moreover, there were two kinds of outliers unique to $PRI_{satellite}$. The first was the striping outliers, which might be caused by the combination of the position of detectors on SGLI and the nearest neighbor sampling method for geometric correction. The second was the cluster outliers, which might be caused by the cluster outliers in VN06. The two kinds of outliers may account for the lower accuracy of $PRI_{satellite}$ in comparison to $CCI_{satellite}$.

In addition to the outliers, small noise also affected the accuracy of $PRI_{satellite}$ more significantly than $CCI_{satellite}$, because of the smaller range of seasonal variation in $PRI_{satellite}$. For vegetation monitoring with satellite remote sensing techniques, small noise may be unavoidable, so we should consider the "susceptibility" of the vegetation index to small noise as well as the ability to represent the target phenomena.

In the future, we should develop a method to remove or correct the outliers and continue to validate the accuracy of $PRI_{satellite}$ and $CCI_{satellite}$ in other vegetation types: paddy field, cropland, and grassland, for instance, as partially investigated by Bayarsaikhan et al. [64]. Furthermore, we need to assess the accuracy over enough footprints; at least one pixel of RSRF products of GCOM-C/SGLI. For the application of $PRI_{satellite}$ and $CCI_{satellite}$, we plan to validate the their ability to track photosynthetic activities by comparing them with in situ eco-physiological data, such as LUE and GPP. We also pay attention to the availability of the green-red vegetation index (GRVI) [56] derived from other satellites, such as MODIS, Sentinel-2, and Himawari-8. The GRVI does not require "narrow" green band reflectance, and Yin et al. [65] have reported that GRVI performed similarly to CCI. Such combinations with eco-physiological data and collaborations over the satellites must provide a better understanding of terrestrial ecosystems with PRI and CCI.

**Author Contributions:** Conceptualization, T.S., T.K.A. and K.N.N.; methodology, T.S. and T.K.A.; software, T.S.; formal analysis, T.S.; field work and data curation, T.S., T.K.A., R.I., K.T., S.T., T.N. and K.N.N.; writing—original draft preparation, T.S.; writing—review and editing, T.S., T.K.A., R.I., S.T. and K.N.N.; visualization, T.S.; supervision, T.S., T.K.A. and K.N.N.; project administration, K.N.N.; funding acquisition, T.S., T.K.A., T.N. and K.N.N. All authors have read and agreed to the published version of the manuscript.

**Funding:** This research was supported by Japan Science and Technology Agency (JST) Support for Pioneering Research Initiated by the Next Generation (SPRING) Grant Numbers JPMJSP2124; Japan Aerospace Exploration Agency (JAXA) Global Change Observation Mission (GCOM) Grant Numbers ER2GCF103 and ER3GCF104; Japan Society for the Promotion of Science (JSPS) KAKENHI Grant Numbers JP19K20433 and JP22H05739

**Data Availability Statement:** The data used in this study are publicly available from the following websites with a user account or a user request: GCOM-C/SGLI data: https://gportal.jaxa.jp/gpr/?lang=en (accessed on 23 August 2022) (see Section 2.3.1) and in situ observation data: http://www.pheno-eye.org/ (accessed on 23 August 2022). The satellite data processing tools are available via https://github.com/tigersasagawa/sgli-tools (accessed on 23 August 2022).

**Acknowledgments:** The authors thank the Japan Aerospace Exploration Agency (JAXA) for providing the satellite data, the Phenological Eyes Network (PEN) for providing in situ data, and Murakami, H. (JAXA); Kobayashi, T. (University of Tsukuba: UT); Phan, C.D. (UT); Iwao, K. (National Institute of Advanced Industrial Science and Technology: AIST); Tsuchida, S. (AIST); and Mizuochi, H. (AIST) for their advice on this study.

**Conflicts of Interest:** The authors declare no conflict of interest.

## Appendix A

**Table A1.** The center wavelengths of the MS-700 bands which were the nearest or the second nearest to 531 nm, 570 nm, and 645 nm and used for the liner interpolation.

| Site ID and Period | Direction of MS-700 | The Nearest Neighbor Band's Peak to 531 nm | The Second Nearest Neighbor Band's Peak to 531 nm | The Nearest Neighbor Band's Peak to 570 nm | The Second Nearest Neighbor Band's Peak to 570 nm | The Nearest Neighbor Band's Peak to 645 nm | The Second Nearest Neighbor Band's Peak to 645 nm |
|---|---|---|---|---|---|---|---|
| TSE 2018-01-01 – 2020-12-31 | upward | 529.70 nm | 533.05 nm | 569.93 nm | 573.28 nm | 643.51 nm | 646.84 nm |
| | downward | 531.35 nm | 528.00 nm | 571.52 nm | 568.18 nm | 645.12 nm | 641.78 nm |
| TKY 2018-01-01 – 2019-05-07 | upward | 529.64 nm | 532.94 nm | 569.14 nm | 572.42 nm | 644.65 nm | 647.93 nm |
| | downward | 529.64 nm | 532.94 nm | 569.14 nm | 572.42 nm | 644.65 nm | 647.93 nm |
| TKY 2019-05-08 – 2020-12-31 | upward | 529.88 nm | 533.17 nm | 569.33 nm | 572.62 nm | 644.93 nm | 648.22 nm |
| | downward | 529.88 nm | 533.17 nm | 569.33 nm | 572.62 nm | 644.93 nm | 648.22 nm |
| FJY 2018-01-01 – 2020-12-31 | upward | 529.85 nm | 533.15 nm | 569.40 nm | 572.69 nm | 645.04 nm | 641.76 nm |
| | downward | 529.76 nm | 533.06 nm | 569.34 nm | 572.64 nm | 644.99 nm | 648.28 nm |
| FHK 2018-01-01 – 2018-12-31 | upward | 532.47 nm | 529.18 nm | 568.65 nm | 571.94 nm | 644.19 nm | 647.47 nm |
| | downward | 530.03 nm | 533.33 nm | 569.54 nm | 572.83 nm | 645.08 nm | 641.81 nm |
| FHK 2019-01-01 – 2020-12-31 | upward | 531.79 nm | 528.45 nm | 568.58 nm | 571.92 nm | 645.43 nm | 642.09 nm |
| | downward | 531.86 nm | 528.51 nm | 568.69 nm | 572.04 nm | 645.69 nm | 642.34 nm |

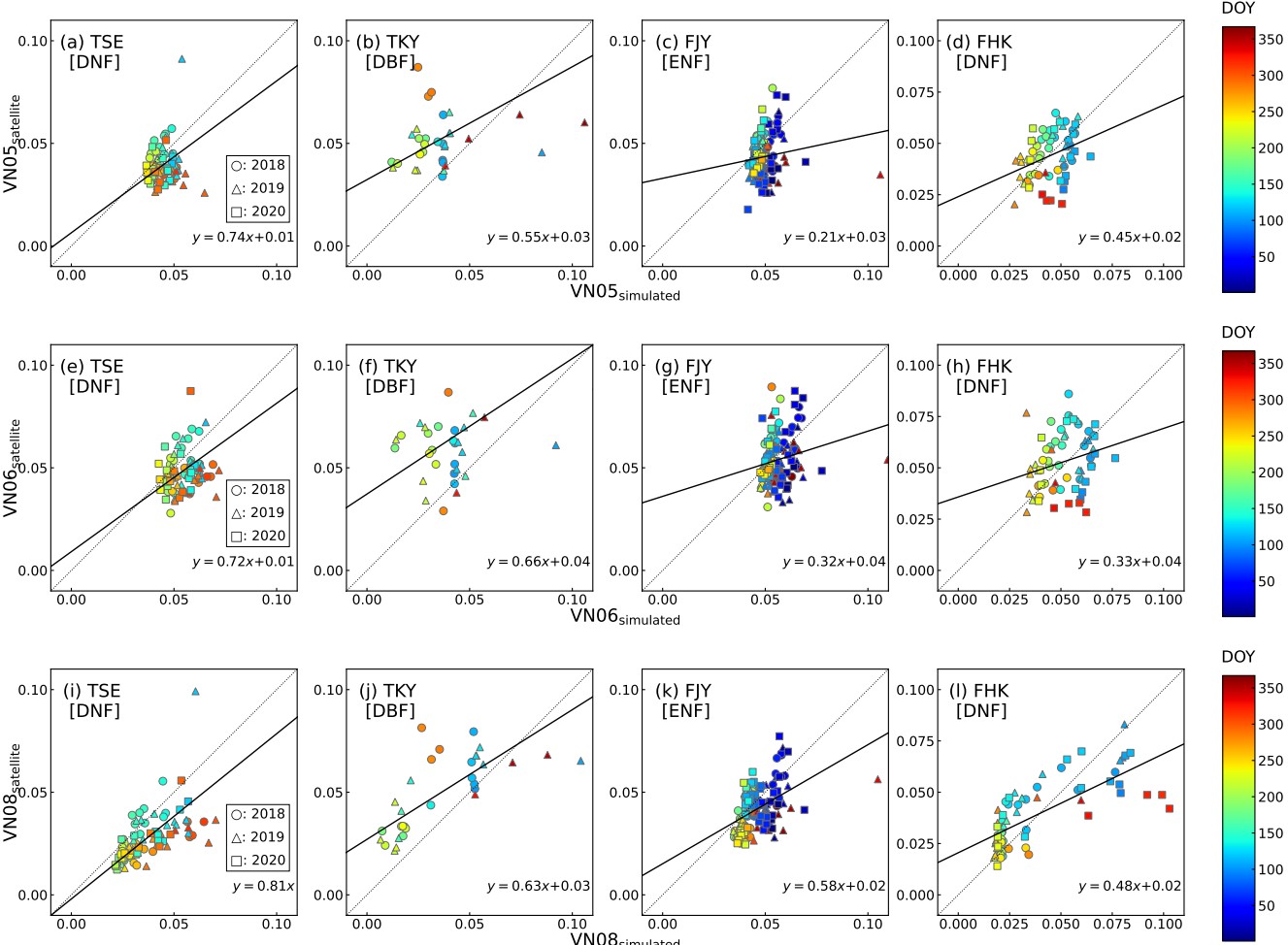

**Figure A1.** The scatter plots of simulated band value and satellite band value used for calculation of PRI and CCI. The shape of each point represents the year. The color of each point corresponds to the DOY. Note that the range of the x-axis and y-axis is limited from 0 to 0.1, so a few points out of the range are not displayed.

**Table A2.** The statistics of the accuracy assessment of VN05, VN06, and VN08.

|  | Site ID | *n* | *r* | *RMSE* | *MAE* |
|---|---|---|---|---|---|
| VN05 | TSE | 84 | 0.856 ($p < 0.001$) | 0.016 | 0.010 |
|  | TKY | 40 | 0.839 ($p < 0.001$) | 0.032 | 0.026 |
|  | FJY | 146 | 0.136 ($p = 0.101$) | 4.948 | 4.908 |
|  | FHK | 65 | 0.343 ($p < 0.01$) | 0.012 | 0.010 |
| VN05 | TSE | 84 | 0.831 ($p < 0.001$) | 0.017 | 0.012 |
|  | TKY | 40 | 0.794 ($p < 0.001$) | 0.037 | 0.030 |
|  | FJY | 146 | 0.199 ($p < 0.05$) | 5.594 | 5.550 |
|  | FHK | 65 | 0.250 ($p < 0.05$) | 0.015 | 0.013 |
| VN05 | TSE | 84 | 0.882 ($p < 0.001$) | 0.019 | 0.014 |
|  | TKY | 40 | 0.899 ($p < 0.001$) | 0.031 | 0.024 |
|  | FJY | 146 | 0.481 ($p < 0.001$) | 4.697 | 4.621 |
|  | FHK | 65 | 0.738 ($p < 0.001$) | 0.017 | 0.013 |

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
