# Peer review of "Accuracy Assessment of Photochemical Reflectance Index (PRI) and Chlorophyll Carotenoid Index (CCI) Derived from GCOM-C/SGLI with In Situ Data"

_remotesensing, doi:10.3390/rs14215352_

Round 1

Reviewer 1 Report

Dear Authors,

Fırst of all, thank you for this interesting paper. I would like to express couple of ideas for the paper.

1. As you know, "all measurements, on or close to Earth surface, are subject to the errors at least due to the gravity. In your case there are many sources to affect your measurements like all the environmental condition such as snow, cloud cover (as you mentioned). I wonder, what would be the results, if you run a parameter estimation method (e.g. least square) and use the result of that analysis as field measurements.

2. Although the spatial resolution of the satellite images are more than enough to cover the sampling sites, as you represented in figure 6, none of the measurement sites is in the center of the pixel (some of them are on the edge of the pixel). I would kindly suggest to include the adjacent pixels to the analysis (e.g. mean/median of the 9 neighbour pixels).

3. PRI uses certain values within the spectrum. There may not much to improve the measurements and the results. However, as you mentioned, CCI can be calculated using an interval  (red wavelength). For the calculation of the CCI, you used MODIS bands 11 and 14. Have use considered to apply the spectral curve of the MODIS sensor to the spectroradiometer measurements to have similar data types for comparisons.

Last of all, please check line 301 for possible typos.

Thank you

Reviewer 2 Report

The authors compared satellite PRI and CCI derived from SGLI to tower based indices at four sites in Japan. Their results showed that satellite CCI achieved a better coherence with ground measurements than PRI. However, both indices are confounded by factors such as georeferencing, cloud coverage, etc. Considering the ‘loss’ of MODIS, SGLI offers the opportunity to provide global CCI and PRI products till the next generation hyperspectral satellite data is available.  The current manuscript reads generally well but more like a technical report. It would be more interesting to see if the authors can dig into the problems they listed. 

More information is required in the methods section. For example, how was the ground data aggregated when compared to the satellite products? Is it the daily data or data collected at the time when the satellite overpasses?

Besides the indices, it might be also good to show the differences in the actual bands between the ground and satellite data. 

What about the effects of difference in footprint between the satellite and ground sensors? 

The authors discussed the cloud coverage and snow effects very briefly. It might be good to dig into this a bit more. Also, PRI and CCI are sensitive to the snow cover. I’m interested in seeing a comparison focusing on the snow-free dataset. 

What about the data availability? Is there a tool that authors can deliver for data extraction so that other people can also use and test this PRI/CCI product?

L10-12: this sentence is a bit vague. Could you please consider rewriting?

L106-107: how the errors were corrected? Please specify. 

L107-109: how often was the MS-700 motor rotating the mirrors?

L155: a little bit more information about the SGLI would be appreciated, for example, the spatial resolution, center of the bands, spectral resolution of each band, revisit time, etc. 

Figure 6: why was the Satinel-2 data used as the background image? 

L188-190: these equations may not be needed here, because they should be straightforward to typical readers. 

Figure 7 (and Figure 9): this figure is useful but might be too busy in the current form. It might be better to just show PRIsimulated and PRIsatellite here. And use another sets of figures to show the relationship between PRIoriginal and PRIsimulated; PRIsatellite(QA!=2) and PRIsatellite(QA=2). The second sets of figures can be included in the supplemental materials. 

Figures 9 and 10: it seems that a lot of 2020 data was missing at TKY, but this was not shown in the PRI figures. 

L272-288: Okay, is there a way to correct for this then?

L289-L296: what may have caused these outliers? 

L297: what does ‘fluctuations’ mean?

Reviewer 3 Report

The general concept of the study may be interesting, since it compares parameters as estimated at different scales, but the presentation of the facts sometimes lacks the clarity. Too much attention is given to outlied data, which in fact originate from errors in measurements. And discussion is too much built on fact coming from the errors in measurements. The fact should be reconsidered and described better in many cases. More specific comments are following:

In abstract I do not understand the sentence “The difference in the agreement between the satellite PRI and CCI must be owing to the outliers, which uniquely occurred in the satellite PRI and the higher robustness of the satellite CCI against fluctuations.” – sentence look to me that PRI and CCI are compared, but I do not understand the role of outliers, and then what is the term “robustness against fluctuations”. But the entire information looks weird. Later it looks to me that PRI are compared to PRI and CCI to CCI, which is correct.

Line 26 – the changes are closely related to the photosynthetic activity.

Line 31 – rather “at” leaf scale

Line 41 – I would delete “instead of that at 570nm as”

Line 51 – was needed

Line 139 – then by the meaning of the sentence the step of data acquisition was 3.3 nm, and “true” value was interpolated based on some two neighboring bands, which were at what position? By spectral resolution we normally understand FWHM.

Line 156 – I do not understand “one day product”

Line 196 – Leafy season is not common phrase

Lines 200-213 – why is so much attention given to some outlied data, which originate most likely from bias in the measurements, and sun elevation or with inaccuracy in spatial assessment.

Figure 8 There is quite small amount of points considering that satellite measurements are running every 2 days. That may be the biggest weakeness of the article, that the trends in data do not look very conclusive. But the dependencies between the same variables from different sources are visible in the CCI data. Maybe all years could fit into one graph.

Line 226 – you mean higher wavelength was used for calculating CCI in one case. This probably require rephrasing.

Line 233 – how can one variable from one source overestimate or underestimate the same variable from different source. The entire concept of these repeating parts requires revision.

From what I see the biggest differences in CCI are between broadleaved and needle leaved species. Is this somewhere discussed, it may be related to LAI and species.

Line 246 – considering the fact that we are comparing variables at different spatial scale “accuracy validation” may not be the relationship we can address. I somehow missed the information what was the foliar area measured using spectrometer on tower. And approximate distance of this sensor from the forest.

Line 247 – was fairly matched “with” maybe

Line 250 – why is number of outliers so important again. Further associating outliers with clouds look weak and insufficient later. This information should be rather given at once.

Line 298 – how is geometric correction different from stripping accuracy? It look to me these are the same information.

Round 2

Reviewer 2 Report

I appreciate the authors' efforts to address my comments and concerns. Congratulations on a work well done! 

Reviewer 3 Report

I read the review response letter and I have gone through the manuscript, and the adjustments look good to me. I believe other reviewers also had many comments, so the manuscript was adjusted also according to these comments. I agree with accepting the manuscript.